# Causal Discovery via Bayesian Optimization

**Bao Duong, Sunil Gupta, and Thin Nguyen**
Applied Artificial Intelligence Institute (A$^2$I$^2$), Deakin University, Geelong, Australia
{b.duong,sunil.gupta,thin.nguyen}@deakin.edu.au

## Abstract

Existing score-based methods for directed acyclic graph (DAG) learning from observational data struggle to recover the causal graph accurately and sample-efficiently. To overcome this, in this study, we propose **DrBO** (**D**AG **r**ecovery via **B**ayesian **O**ptimization)—a novel DAG learning framework leveraging Bayesian optimization (BO) to find high-scoring DAGs. We show that, by sophisticatedly choosing the promising DAGs to explore, we can find higher-scoring ones much more efficiently. To address the scalability issues of conventional BO in DAG learning, we replace Gaussian Processes commonly employed in BO with dropout neural networks, trained in a continual manner, which allows for (i) flexibly modeling the DAG scores without overfitting, (ii) incorporation of uncertainty into the estimated scores, and (iii) scaling with the number of evaluations. As a result, **DrBO** is computationally efficient and can find the accurate DAG in fewer trials and less time than existing state-of-the-art methods. This is demonstrated through an extensive set of empirical evaluations on many challenging settings with both synthetic and real data. Our implementation is available at https://github.com/baosws/DrBO.

## 1 Introduction

Learning directed acyclic graphs (DAGs) encoding the underlying causal relationships, also known as causal discovery, provides invaluable insights about interventional outcomes and counterfactuals, and thus significant research effort has been dedicated on this frontier. Our study focuses on the score-based framework—a major class of causal discovery methods which casts the DAG learning problem as an optimization problem, maximizing over the space of DAGs a predefined score function measuring how a DAG $\mathcal{G}$ fits the observed data $\mathcal{D}$:

$$\mathcal{G}^* = \arg\max_{\mathcal{G} \in \text{DAGs}} S\left(\mathcal{D}, \mathcal{G}\right). \tag{1}$$

There are several challenges associated with this formulation. First, this optimization problem is NP-hard in general (Chickering, 1996), due to the combinatorial search domain that scales super-exponentially with the graph size (Robinson, 1977) and the acyclicity condition that is nontrivial to maintain. The second challenge is computational cost, as the score evaluation can be expensive for complex models (Zhu et al., 2020; Wang et al., 2021), and hence, methods requiring too many trials will incur a heavy computational expense.

Bayesian Optimization (BO) has emerged as an effective approach for expensive black-box optimization thanks to its sample efficiency. Its applicability covers many domains due to the pervasiveness of optimization tasks in virtually every field. The central idea is that past evaluation data can reveal potential candidates to evaluate next, and effectively exploiting them allows us to arrive at better solutions using fewer evaluations. However, while BO has been employed for *active causal discovery* (Toth et al., 2022; Zhang et al., 2024) to suggest cost-effective intervention strategies to discover causal graphs from active experiments, its power is yet to be harnessed for the problem of *observational causal discovery*, where no active intervention is available.

We have identified two potential difficulties preventing BO to optimize Eq. (1). The first is the limited scalability of BO in general, which is usually restricted to only a few hundred dimensions and thousands of evaluations (Wang et al., 2023), while DAG learning often involves far more trials (Zhu et al., 2020; Wang et al., 2021; Duong et al., 2025). The second challenge is how to efficiently optimize the "acquisition function" measuring the potential of DAG solutions in BO. This itself is a score-based DAG learning problem and is expected to be cheaper than the original problem as it is repeated many times in the BO pipeline, thus requiring to be very efficient to be practical.

**Present study.** In this study, we tackle these obstacles and bring forth the benefits of BO into causal discovery with the introduction of **DrBO** (**D**AG **r**ecovery via **B**ayesian **O**ptimization)—a novel causal discovery algorithm that leverages BO to find the highest-scoring DAG. More particularly, we solve the aforementioned challenges by several key design choices. (i) Inspired by Yu et al. (2021); Massidda et al. (2024); Duong et al. (2025), we devise a low-rank DAG representation that relaxes the constrained optimization in Eq. (1) to an *unconstrained optimization problem* with a dimensionality growing linearly with the number of nodes (Sec. 4.1), enabling the use of BO with an amenable dimensionality. (ii) We replace the Gaussian processes (GPs) in conventional BO approaches that scale cubically w.r.t. the number of evaluations with *dropout neural networks* (Srivastava et al., 2014; Guo et al., 2021), which offer expressive uncertainty-aware modeling capabilities, as well as faster acquisition function calculation and optimization (Sec. 4.3). (iii) Our surrogate model *learns the DAG score indirectly* via node-wise local scores, allowing us to predict the DAG scores more accurately with enhanced training information, compared with learning a direct map from a DAG to its score and not exploiting the local scores (Sec. 4.4). (iv) Instead of being retrained every step with all data, our neural networks are *trained continually*, enabling our method to scale better with the number of trials (Sec. 4.5). Compared to existing optimization strategies in causal discovery, like gradient-based methods which do not attempt to exploit past exploration data to prioritize visiting the most promising DAGs, our method makes more informed decisions about which DAGs to investigate next, leading to fewer unnecessary trials to reach high-scoring DAGs.

**Contributions.** The main contributions of our study are summarized as follows:

1. To facilitate sample-efficient causal discovery, we propose **DrBO**—a causal discovery method employing BO to optimize for the DAG score. **DrBO** is specifically designed for causal discovery, aiming to be not only accurate, but also computationally manageable. *To our knowledge, this is the first score-based causal discovery method based on BO for purely observational data.*

2. We demonstrate the effectiveness of our method on a comprehensive set of experiments, showing that **DrBO** can consistently surpass state-of-the-art baselines on various conditions, with fewer evaluations and less time, signifying the sample efficiency of BO in our design. In addition, extensive ablation studies verify the significance of our design choices.

## 2    RELATED WORK

Causal discovery methods can be broadly categorized into two major classes, namely constraint-based and score-based methods. The former category (Spirtes et al., 2000; Colombo et al., 2012) involves performing a series of hypothesis tests to recover the undirected skeleton of the causal graph, before orienting the edges using graphical rules. On the other hand, score-based causal discovery recasts the problem as a combinatorial optimization task. Classical methods (Chickering, 2002; Ramsey et al., 2017; Ramsey, 2015) greedily traverse the DAG space by adding and removing edges one-at-a-time to maintain acyclicity. Recent advances include relaxing the combinatorial optimization to a continuous optimization problem (Zheng et al., 2018; Yu et al., 2019; Zheng et al., 2020; Yu et al., 2021; Zhang et al., 2022; Bello et al., 2022; Annadani et al., 2023). In addition, methods based on Reinforcement learning (RL) (Zhu et al., 2020; Wang et al., 2021; Yang et al., 2023a; Duong et al., 2025) have recently emerged as competitive search strategies. We also acknowledge *interventional causal discovery* studies (Hauser & Bühlmann, 2012; Brouillard et al., 2020; Lippe et al., 2022) where interventional data is exploited to help identify the causal DAG, however, here we focus on the more challenging setting where no intervention is available.

Furthermore, *Bayesian causal discovery* studies are an intriguing direction (Deleu et al., 2022; Tran et al., 2023; Annadani et al., 2023), where the aim is to infer the posterior distribution over causal graphs given observed data, in order to quantify uncertainty in *DAG estimates*. Meanwhile, our use of BO involves quantifying uncertainty in *DAG scores*, so despite sharing the term "Bayesian", these studies are distant from us. Moreover, while Bayesian optimization has been employed for causal discovery in the *active* setting (Toth et al., 2022; Zhang et al., 2024), these methods utilize BO to suggest optimal interventions to quickly recover the causal DAG, necessitating the ability to perform active experiments. Meanwhile, our study utilizes BO to optimize for the score function that can be calculated purely from observational data. In addition, *causal Bayesian optimization* (Aglietti et al., 2020; 2021), which deals with optimizing a variable of interest that is part of a causal system with *known DAG* via a series of interventions suggested by BO, is also a different problem from ours, which focuses on finding the unknown DAG instead.

## 3 BACKGROUND

**Notations.** In this paper, unless specifically indicated, normal lowercase letters indicate scalars (e.g., $x$, $y$) or functions (e.g., $f$, $g$), while bold lowercase letters represent vectors (e.g., $\mathbf{x}$, $\mathbf{y}$), and bold uppercase letters denote matrices (e.g., $\mathbf{X}$). Meanwhile, subscripts and bracketed superscripts index dimensions and samples, respectively, e.g., $x_i^{(j)}$ denotes the $i$-th dimension of the $j$-th sample.

### 3.1 BAYESIAN OPTIMIZATION

We provide here only the details necessary to understand our contributions. For a more comprehensive review of BO, see Wang et al. (2023), for example. Consider a maximization of a function $f$ that is expensive to evaluate: $\mathbf{x}^* = \arg\max_{\mathbf{x} \in \mathcal{X}} f(\mathbf{x})$. BO is a class of sequential optimization methods, which iteratively (i) proposes potential candidate(s) to evaluate based on an "acquisition function", then (ii) evaluates said candidate(s), and (iii) updates its statistical model with the newly acquired observations. More specifically, BO defines a probabilistic "surrogate" model over the distribution $f$ given observed data $\mathbf{X}, \mathbf{y}$, i.e., $P(f \mid \mathbf{X}, \mathbf{y})$, to define the acquisition function.

**Surrogate model.** To model $P(f \mid \mathbf{X}, \mathbf{y})$, Gaussian process (GP) is the standard in BO since it offers a closed-form solution (Rasmussen, 2003). Assuming we have observed a dataset of $n$ evaluations $\mathbf{X} = \left[\mathbf{x}^{(1)}, \ldots, \mathbf{x}^{(n)}\right]^\top$ and $\mathbf{y} = \left[y^{(1)}, \ldots, y^{(n)}\right]^\top$, where $\mathbf{x} \sim \mathcal{X} = \mathbb{R}^d$ and $y := f(\mathbf{x})$, GP assumes that $\mathbf{y}$ follows a multivariate Gaussian distribution governed by a mean function $\mu : \mathcal{X} \to \mathbb{R}$ and positive-definite covariance function $\kappa : \mathcal{X} \times \mathcal{X} \to \mathbb{R} : \mathbf{y} \sim \mathcal{N}(\mu(\mathbf{x}), \mathbf{K}_{\mathbf{X},\mathbf{X}})$ where $\mathbf{K}_{\mathbf{X},\mathbf{X}} := \left[\kappa\left(\mathbf{x}^{(i)}, \mathbf{x}^{(j)}\right)\right]_{i,j=1,\ldots,n}$ is the $n \times n$ covariance matrix. Using Bayes' theorem, the posterior of the function value at a new location $\mathbf{x}$ is given analytically as: $P(y \mid \mathbf{x}, \mathbf{X}, \mathbf{y}) = \mathcal{N}\left(\mu(\mathbf{x}), \sigma^2(\mathbf{x})\right)$ where $\mu(\mathbf{x}) := \mathbf{k}_{\mathbf{x},\mathbf{X}} \mathbf{K}_{\mathbf{X},\mathbf{X}}^{-1} \mathbf{y}$ and $\sigma^2(\mathbf{x}) := \kappa(\mathbf{x}, \mathbf{x}) - \mathbf{k}_{\mathbf{x},\mathbf{X}}^\top \mathbf{K}_{\mathbf{X},\mathbf{X}}^{-1} \mathbf{k}_{\mathbf{x},\mathbf{X}}$. GPs scale poorly with $n$ due to the need to invert $\mathbf{K}_{\mathbf{X},\mathbf{X}}$, so alternative statistical models like random forest (Hutter et al., 2011), Bayesian linear regression (Snoek et al., 2015), and Bayesian neural network (Springenberg et al., 2016) have been employed as more scalable surrogate models.

**Acquisition functions** (AFs) in BO judge how promising an arbitrary candidate $\mathbf{x}$ is based on the posterior inferred by the surrogate model, to make a more informed candidate suggestion. As an example, upper confidence bound (UCB) is a common choice, designed to minimize regret in the multi-armed bandit literature (Srinivas et al., 2010) and given by $\mathrm{AF}(\mathbf{x}) := \mu(\mathbf{x}) + \beta\sigma(\mathbf{x})$, where $\beta > 0$ is a hyperparameter controlling the exploitation-exploration trade-off. Another promising acquisition is Thompson sampling (TS, Thompson, 1933), which is a stochastic function that uses a random draw from the posterior as the potential indicator: $\mathrm{AF}(\mathbf{x}) \sim P(y \mid \mathbf{x}, \mathbf{X}, \mathbf{y})$.

### 3.2 STRUCTURAL CAUSAL MODEL

Let $\mathbf{x} \in \mathbb{R}^d$ be the random vector capturing the system of interest, and $\mathcal{D} = \left\{\mathbf{x}^{(k)}\right\}_{k=1}^n$ denote an i.i.d. dataset of $n$ samples from $P(\mathbf{x})$. The structural causal model (SCM, Pearl, 2000; 2009) among said variables can be described by (i) a DAG $\mathcal{G} = (\mathcal{V}, \mathcal{E})$ where each node $i \in \mathcal{V} = \{1, \ldots, d\}$ corresponds to a random variable $x_i$, and each edge $(j \to i) \in \mathcal{E} \subset \mathcal{V} \times \mathcal{V}$ implies that $x_j$ is a direct cause of $x_i$, (ii) a set of functions $\{f_i\}_{i=1}^n$ dictating the causal mechanisms, and (iii) a noise distribution $P(\boldsymbol{\varepsilon})$. Together, these components define a generative process $x_i := f_i\left(\mathbf{x}_{\mathrm{pa}_i^{\mathcal{G}}}, \varepsilon_i\right), \forall i = 1, \ldots, d$, where $\mathrm{pa}_i^{\mathcal{G}} = \{j \in \mathcal{V} \mid (j \to i) \in \mathcal{E}\}$ is the set of direct causes (a.k.a. structural parents) of node $i$ in $\mathcal{G}$, and $\boldsymbol{\varepsilon} \sim P(\boldsymbol{\varepsilon})$ is the noise vector. Then, the observational causal discovery problem is concerned about recovering the DAG $\mathcal{G}$ from the observational dataset $\mathcal{D}$. In addition, following (Zhu et al., 2020; Wang et al., 2021; Yang et al., 2023a;b; Duong et al., 2025), we also assume: (i) *causal sufficiency*: there is no unobserved confounders among the variables; (ii) *causal minimality*: there is no function $f_i$ that is constant to any of its argument (Peters et al., 2014); and (iii) *identifiable causal model*s: this means $\mathcal{G}$ is the unique causal graph that can induce $P(\mathbf{x})$, and thus it is possible to be recovered. For instance, while general linear-Gaussian models are known to be unidentifiable (Spirtes et al., 2000), examples for identifiable causal models include linear-Gaussian models with the equal-variance assumption (Peters et al., 2014) and nonlinear additive noise models (ANMs) in general (Hoyer et al., 2008). Our experiments will adopt these identifiable models.

### 3.3 SCORE-BASED CAUSAL DISCOVERY

A critical component of score-based causal discovery is the proper specification of a scoring function. With the proper scoring function, the optimization of the score is equivalent to reaching to

ground truth DAG. Consistent scoring functions (Chickering, 2002) are known to satisfy this requirement. In the main text, we demonstrate our method with the Bayesian Information Criterion (BIC, Schwarz, 1978), which is a consistent score as shown by Haughton (1988) and is widely considered in numerous existing studies (Chickering, 2002; Zhu et al., 2020; Wang et al., 2021; Yang et al., 2023a;b; Duong et al., 2025).

More formally, let $\theta := \left\{ \{f_i\}_{i=1}^d, P(\varepsilon) \right\}$ be the parameters of an SCM, then the BIC score is given by $S_{\text{BIC}}(\mathcal{D}, \mathcal{G}) := 2 \ln p\left(\mathcal{D} \mid \hat{\theta}, \mathcal{G}\right) - |\mathcal{G}| \ln n$, where $\hat{\theta} := \arg\max_\theta p(\mathcal{D} \mid \theta, \mathcal{G})$ is the maximum-likelihood estimator of the causal model parameters, $n$ is the sample size of $\mathcal{D}$, and $|\mathcal{G}|$ denotes the number of edges in $\mathcal{G}$. The generality of BIC allows for its adaptation to numerous causal models. In the main paper, we showcase our method with the BIC defined for the popular additive noise model (ANM, Hoyer et al., 2008): $x_i := f_i\left(\mathbf{x}_{\text{pa}_i}\right) + \varepsilon_i$, where $\varepsilon_i \sim \mathcal{N}\left(0, \sigma_i^2\right)$. In the general form, where the noise variances can be non-equal, the BIC for ANM is specified as follows:

$$S_{\text{BIC-NV}}(\mathcal{D}, \mathcal{G}) := -n \sum_{i=1}^d \ln \text{MSE}_i\left(\text{pa}_i^{\mathcal{G}}\right) - |\mathcal{G}| \ln n, \tag{2}$$

where NV stands for "non-equal variance" and $\text{MSE}_i\left(\text{pa}_i^{\mathcal{G}}\right) := \frac{1}{n} \sum_{j=1}^n \left(x_i^{(j)} - \hat{f}_i\left(\mathbf{x}_{\text{pa}_i^{\mathcal{G}}}^{(j)}\right)\right)^2$ is the mean squared error after regressing $x_i$ on $\mathbf{x}_{\text{pa}_i^{\mathcal{G}}}$. In addition, if we further assume that the noise variables have equal variances (Bühlmann et al., 2014) then the BIC yields:

$$S_{\text{BIC-EV}}(\mathcal{D}, \mathcal{G}) := -nd \ln \frac{\sum_{i=1}^d \text{MSE}_i\left(\text{pa}_i^{\mathcal{G}}\right)}{d} - |\mathcal{G}| \ln n. \tag{3}$$

Eqs. (2) and (3) are common in prior studies (Zhu et al., 2020; Wang et al., 2021; Yang et al., 2023a;b; Duong et al., 2025), and we also provide their derivation in Appendix B.

### 3.4 PARAMETRIZED DAG GENERATION

For effective acquisition function optimization, we find it crucial for our method to be able to quickly generate candidate DAGs within specific regions determined by a low-dimensional search space. This would help narrow down the regions of interest and improve the quality of suggested candidates. A potential approach towards this end is autoregressive DAG generation techniques (Wang et al., 2021; Deleu et al., 2022; Yang et al., 2023a; Deleu et al., 2024), which break down the DAG generation into multiple sequential steps. However, each step of the generation process is computationally involved due to their autoregressive nature (Duong et al., 2025), accumulating into a significant DAG generation cost. Recently, constraint-free DAG representations with a quadratic complexity are proposed in (Yu et al., 2021; Massidda et al., 2024; Duong et al., 2025), which introduce different maps from an unconstrained real-valued representation to the space of DAGs. For example, the Vec2DAG operator in (Duong et al., 2025) takes as input a continuous "node potential" vector $\mathbf{p} \in \mathbb{R}^d$ and strictly upper-triangular "edge potential" matrix $\mathbf{E} \in \mathbb{R}^{d \times d}$ to deterministically create a DAG: $\text{Vec2DAG}(\mathbf{p}, \mathbf{E}) := H(\text{grad}(\mathbf{p})) \odot H\left(\mathbf{E} + \mathbf{E}^\top\right)$, where $H(x) := \begin{cases} 1, & \text{if } x > 0, \\ 0, & \text{otherwise} \end{cases}$ is the entry-wise Heaviside step function, $\odot$ is the Hadamard product, and $\text{grad}(\mathbf{p})_{ij} := p_j - p_i$ is the gradient flow operator (Lim, 2020).

**Example.** Consider a system of 3 nodes with potentials $\mathbf{p} = [-1, 3, 2]$ and $\mathbf{E} = \begin{bmatrix} 0 & 2 & -4 \\ 0 & 0 & 7 \\ 0 & 0 & 0 \end{bmatrix}$.

Then, $\text{Vec2DAG}(\mathbf{p}, \mathbf{E}) = H\left(\begin{bmatrix} 0 & 4 & 3 \\ -4 & 0 & -1 \\ -3 & 1 & 0 \end{bmatrix}\right) \odot H\left(\begin{bmatrix} 0 & 2 & -4 \\ 2 & 0 & 7 \\ -4 & 7 & 0 \end{bmatrix}\right) = \begin{bmatrix} 0 & 1 & 0 \\ 0 & 0 & 0 \\ 0 & 1 & 0 \end{bmatrix}$.

This is the adjacency of the DAG $1 \to 2 \leftarrow 3$, where the edge directions and connectivities are determined by the first and second terms of Vec2DAG, respectively.

Using this approach, it is simple to sample candidate DAGs whose representations are in the neighborhood around some values $\mathbf{p}$ and $\mathbf{E}$ of interest.

---

**Algorithm 1** The **DrBO** method for causal discovery.

---

**Require:** Dataset $\mathcal{D} = \left\{ \mathbf{x}^{(j)} \in \mathbb{R}^d \right\}_{j=1}^n$ of $d$ nodes and $n$ observations, score function $S\left(\mathcal{D}, \cdot\right)$,
  DAG rank $k$, batch size $B$, no. of preliminary candidates $C$, and total no. of evaluations $T$.
**Ensure:** A DAG $\hat{\mathcal{G}}$ that maximizes $S\left(\mathcal{D}, \mathcal{G}\right)$.

1: Initialize empty experience $\mathcal{H} := \emptyset$ and node-wise dropout neural nets: $\{\text{DropoutNN}_i\}_{i=1}^d$.
2: **while** $|\mathcal{H}| < T$ **do**
3:   Generate random DAGs: $\left\{ \mathcal{G}^{(j)} := \tau\left(\mathbf{z}^{(j)}\right) \right\}_{j=1}^C$ where $\mathbf{z} \in [-1, 1]^{d(1+k)}$. ▷ Secs. 4.1 & 4.2.
4:   Sample local scores: $\left\{ \left\{ l_i^{(j)} \sim \text{DropoutNN}_i\left(\text{pa}_i^{\mathcal{G}^{(j)}}\right) \right\}_{i=1}^d \right\}_{j=1}^C$. ▷ Sec. 4.3.
5:   Combine local scores: $\left\{ \text{AF}^{(j)} := \text{Combine}\left(l_1^{(j)}, \dots, l_d^{(j)}\right) \right\}_{j=1}^C$. ▷ Sec. 4.4.
6:   Select top $B$ candidates with highest AF values: $j_1, \dots, j_B := \underset{j=1,\dots,C}{\text{argtop}_B} \text{AF}^{(j)}$. ▷ Sec. 4.2.
7:   Evaluate these candidates and update experience: $\mathcal{H} := \mathcal{H} \cup \left\{ \left(\mathcal{G}^{(j)}, S\left(\mathcal{D}, \mathcal{G}^{(j)}\right)\right) \right\}_{j=j_1,\dots,j_B}$.
8:   Update the neural nets on new $\mathcal{H}$. ▷ Sec. 4.5.
9: **end while**
10: Get highest-scoring DAG so far: $\hat{\mathcal{G}} := \arg\max_{\mathcal{G} \in \mathcal{H}} S\left(\mathcal{D}, \mathcal{G}\right)$.
11: Prune $\hat{\mathcal{G}}$ if needed. ▷ Sec. 4.6.

---

## 4  DrBO: DAG RECOVERY VIA BAYESIAN OPTIMIZATION

An overview of our framework is illustrated in Algorithm 1. In the following, we describe the proposed **DrBO** algorithm step-by-step.

### 4.1  SEARCH SPACE

To effectively utilize BO, the search space should be unconstrained. Thus, following Yu et al. (2021); Massidda et al. (2024); Duong et al. (2025), we transform the constrained combinatorial optimization problem in Eq. (1) to an unconstrained optimization task. However, the dimensionality of $\mathcal{O}\left(d^2\right)$ of their search spaces can still be reduced to mitigate the effect of the curse of dimensionality, facilitating easier acquisition function optimization. For this purpose, similarly to Fang et al. (2023), we assume that the edge potential matrix in Vec2DAG (Duong et al., 2025) is low-rank and thus consider an adaptation that offers a search space that grows linearly with the number of nodes. Specifically, each node $i$ is now associated with a low-dimensional embedding vector $r_i \in \mathbb{R}^k$ with $k \ll d$, and two nodes $i$ and $j$ are connected if and only if $\langle r_i, r_j \rangle > 0$. The total dimensionality of this search space is thus only $d \cdot (1 + k)$. More formally, given a node potential $\mathbf{p} \in \mathbb{R}^d$ and an embedding matrix $\mathbf{R} \in \mathbb{R}^{d \times k}$, we define the following map

$$\tau\left(\mathbf{p}, \mathbf{R}\right) := H\left(\text{grad}\left(\mathbf{p}\right)\right) \odot H\left(\mathbf{R} \cdot \mathbf{R}^\top\right). \tag{4}$$

The following Lemma ensures the acyclicity of the DAG corresponding to $\tau\left(\mathbf{p}, \mathbf{R}\right)$.

**Lemma 1.** *For all $d, k \in \mathbb{N}^+$, $\mathbf{p} \in \mathbb{R}^d$ and $\mathbf{R} \in \mathbb{R}^{d \times k}$, let $\tau : \mathbb{R}^d \times \mathbb{R}^{d \times k} \to \{0, 1\}^{d \times d}$ be defined as in Eq. (4). Then, $\tau\left(\mathbf{p}, \mathbf{R}\right)$ represents a binary adjacency matrix of a DAG.*

The proof can be found in Appendix A.1. In addition, like Vec2DAG, our variation also exhibits the following scale-invariance property.

**Lemma 2.** *For all $d, k \in \mathbb{N}^+$, $\mathbf{p} \in \mathbb{R}^d$ and $\mathbf{R} \in \mathbb{R}^{d \times k}$, let $\tau : \mathbb{R}^d \times \mathbb{R}^{d \times k} \to \{0, 1\}^{d \times d}$ be defined as in Eq. (4). Then, for all $\alpha > 0$, $\tau\left(\mathbf{p}, \mathbf{R}\right) = \tau\left(\alpha\mathbf{p}, \alpha\mathbf{R}\right)$.*

The proof is provided in Appendix A.2. This insight allows us to restrict the search domain to a fixed range (e.g., $[-1, 1]$) for numerical stability. For brevity, the remaining parts of this manuscript will use vector $\mathbf{z}$ of $d \cdot (1 + k)$ dimensions as the concatenation of $\mathbf{p}$ and the flattened $\mathbf{R}$, and we adopt the notation $\tau\left(\mathbf{z}\right) \equiv \tau\left(\mathbf{p}, \mathbf{R}\right)$. In short, we can now translate the original optimization problem in Eq. (1) to the following unconstrained optimization problem:

$$\mathbf{z}^* := \underset{\mathbf{z} \in \mathbb{R}^{d(1+k)}}{\arg\max} S\left(\mathcal{D}, \tau\left(\mathbf{z}\right)\right). \tag{5}$$

Obviously, if $k \geq d$ then $\mathbf{R} \cdot \mathbf{R}^\top$ is full-rank, so $\tau$ can represent any DAG possible (Theorem 1, Duong et al., 2025), and thus $\mathcal{G}^* := \tau(\mathbf{z}^*)$ is a maximizer of Eq. (1) for any maximizer $\mathbf{z}^*$ of Eq. (5). While this may not hold for $k < d$, our empirical evaluations reveal that this representation suffices even for very complex graphs. Further, we find that lower ranks typically result in better sample-efficiency than higher ranks (Figure 3(a)) and provide an explanation in Appendix F.1.6.

## 4.2 ACQUISITION FUNCTION OPTIMIZATION

To perform each step of the BO pipeline (Sec. 3.1), we use the acquisition function obtained so far to select a batch of $B$ most promising candidates to evaluate, also known as Batch BO (Joy et al., 2020). This is done by first generating $C \geq B$ preliminary candidates $\left\{\mathbf{z}^{(j)}\right\}_{j=1}^{C}$ from a hypercube centered at the current best solution $\mathbf{z}^*$ (see Appendix C.1 for more details). Subsequently, we evaluate the acquisition function of the DAGs induced by these candidates,[1] and choose the top $B$ candidates based on the acquisition function values. The quality of the candidates batch strongly depends on $C$, i.e., if we can evaluate the acquisition function of a lot of candidates, then the top $B$ candidates are likely to have higher values. However, this also increases computational cost, so our acquisition function must scale well with the number of candidates $C$ to mitigate this overhead, which leads us to the next point.

## 4.3 SURROGATE MODELING WITH DROPOUT NETWORKS

To overcome the scalability issues of standard BO due to the use of GPs as discussed earlier, we instead pursue neural networks, which are well-known for their scalability and flexibility (Snoek et al., 2015). Our networks must be able to model uncertainty to help the optimizer prioritize evaluating uncertain but promising candidates. Towards this end, we employ dropout activations (Srivastava et al., 2014), whose original purpose was to reduce overfitting in training neural networks, and were later found to be also useful as an approximate Bayesian inference method (Gal & Ghahramani, 2016), and thus have been successfully applied to BO (Guo et al., 2021). We provide a detailed discussion on this choice compared with other models in Appendix C.2.

Specifically, we devise a single-layer neural network with dropout activation as follows. Let $p \in (0, 1)$ be the dropout rate, $d$ denotes the dimensionality of the input, $h$ is the number of hidden units, $\mathbf{W}_1 \in \mathbb{R}^{d \times h}$ and $\mathbf{W}_2 \in \mathbb{R}^{h \times 1}$ are weight matrices, $\mathbf{b}_1 \in \mathbb{R}^h$ and $b_2 \in \mathbb{R}$ are biases. Our dropout networks are then defined as:

$$\text{DropoutNN}(\mathbf{x}) := \mathbf{W}_2^\top \left( \text{BatchNorm} \left( \text{ReLU} \left( \frac{1}{1-p} \left( (1 - \mathbf{m}) \circ \left( \mathbf{W}_1^\top \mathbf{x} + \mathbf{b}_1 \right) \right) \right) \right) \right) + b_2,$$

where we also follow common practice to employ Batch Normalization (Ioffe & Szegedy, 2015) for improved training efficiency, and $\mathbf{m} \sim \text{Bernoulli}(p)^h$ is drawn for every invocation of $\text{DropoutNN}(\mathbf{x})$ in both train and test modes. By training this model on observed data $\mathbf{X}$ and $\mathbf{y}$ with the square loss, performing a stochastic forward pass $y \sim \text{DropoutNN}(\mathbf{x})$ can be interpreted as drawing from an approximate posterior $y \sim q(y \mid \mathbf{x}, \mathbf{X}, \mathbf{y})$ (Gal & Ghahramani, 2016). Using Thompson sampling, we do not need to characterize the whole posterior and this mere sample suffices for acquisition function optimization (Russo & Van Roy, 2014; Eriksson et al., 2019).

## 4.4 INDIRECT SURROGATE MODELING

A naïve surrogate modeling approach for DAG learning is modeling a direct map from a DAG to its score. However, DAG scores can typically be decomposed into independent node-wise components, which can be further exploited for better modeling. For example, the BIC scores in Eqs. (2) and (3) involve the local components $\left\{ \text{MSE}_i \left( \text{pa}_i^{\mathcal{G}} \right) \right\}_{i=1}^{d}$ observable after each DAG score invocation. To exploit these information to the fullest, we propose to learn local surrogate models predicting the node-wise scores, then combine them using the rule in Eq. (2) or (3), depending on the situation.

Particularly, for each node $i$, we use the evaluation data $\left\{ \left( \text{pa}_i^{\mathcal{G}^{(j)}}, \text{MSE}_i \left( \text{pa}_i^{\mathcal{G}^{(j)}} \right) \right) \right\}_{j=1}$, where $\mathcal{G}^{(j)}$ is the $j$-th evaluated DAG, to train a dropout network $\text{DropoutNN}_i$ predicting $\ln \text{MSE}_i$ from $\text{pa}_i^{\mathcal{G}}$, which is represented by a binary vector of $d$ dimensions. The models are independent among

---

[1]Our AFs do not take as input $\mathbf{z}$, but $\tau(\mathbf{z})$ instead, as many $\mathbf{z}$'s can produce the same DAG.

all nodes instead of being shared to avoid spurious correlations. To summarize, for a dataset of $d$ nodes, we jointly train $d$ local surrogate models $\{\text{DropoutNN}_i\}_{i=1}^d$. To sample the DAG score of a graph $\mathcal{G}$, we first sample each local score $l_i \sim \text{DropoutNN}_i\left(\text{pa}_i^{\mathcal{G}}\right)$, then combine them with $\text{AF}\left(\mathcal{G}\right) := \text{Combine}_{\text{BIC}-\text{NV}}\left(l_1, \ldots, l_d\right) := -n\sum_{i=1}^d l_i - |\mathcal{G}|\ln n$, if the non-equal variance BIC score is being considered, or $\text{AF}\left(\mathcal{G}\right) := \text{Combine}_{\text{BIC}-\text{EV}}\left(l_1, \ldots, l_d\right) := -nd\ln\frac{\sum_{i=1}^d e^{l_i}}{d} - |\mathcal{G}|\ln n$ for the case of equal variance BIC score, which resemble Eqs. (2) and (3), respectively.

### 4.5 Continual Model Training

Upon acquiring a batch of $B$ of new evaluations, the neural networks require retraining to update their weights. To avoid retraining with all data so far, which scales at least quadratically with the number of evaluations, we adopt a continual training approach (Wang et al., 2024) that only updates the models with the new data and a small portion of past data to mitigate forgetting. Specifically, we perform $n_{\text{grads}}$ gradient steps within each BO iteration, each of which is calculated on the $B$ new datapoints and a replay buffer of $n_{\text{replay}}$ past observations using Reservoir Sampling (Vitter, 1985).

### 4.6 Finalizing the Result

Pruning the resultant DAG is common practice to suppress the redundant edges (Bühlmann et al., 2014; Zheng et al., 2018; Wang et al., 2021; Bello et al., 2022; Duong et al., 2025), and is also employed in our framework. This can be done by thresholding the weight matrix for linear data (Zheng et al., 2018), or employing significance testing for nonlinear data using generalized additive model regression (CAM pruning, Peters et al., 2014), or conditional independence testing under the faithfulness assumption (Duong et al., 2025). More details are provided in Appendix C.3.

## 5 Experiments

In this section, we verify our claim in the introduction: **DrBO** *is both more accurate and sample-efficient than existing approaches in score-based observational DAG learning*. We show this by comparing our **DrBO** method with a number of the most recent advances in causal discovery that are based on sequential optimization, including gradient-based methods DAGMA (Bello et al., 2022), COSMO (Massidda et al., 2024), GOLEM (Ng et al., 2020), NOTEARS (Zheng et al., 2018) with TMPI constraint (Zhang et al., 2022), as well as RL-based approaches CORL (Wang et al., 2021) and ALIAS (Duong et al., 2025). We note that CORL, ALIAS, and **DrBO** directly optimize the BIC score, COSMO, DAGMA, and NOTEARS optimize the penalized least-square loss, while GOLEM optimizes a penalized log-likelihood. For gradient-based methods, we consider a gradient update equivalent to one DAG evaluation. Additional information, including implementation details and metrics, are provided in Appendix D.

### 5.1 Results on Synthetic data

We consider the standard Erdős-Rényi (ER) graph model (Erdős & Rényi, 1960) to generate data, where graphs with $d$ nodes and $de$ edges on average are referred to as $d$ER$e$ graphs (e.g, 10ER4).

#### 5.1.1 Linear-Gaussian data

After simulating a DAG $\mathcal{G}$, we sample edge weights with $w_{ji} \sim \mathcal{U}\left([-2, -0.5] \cup [0.5, 2]\right)$ like prior studies (Zheng et al., 2018; Zhu et al., 2020; Wang et al., 2021). Then, we generate a dataset of $n = 1{,}000$ i.i.d. samples according to a linear-Gaussian SCM $x_i := \sum_{j\in\text{pa}_i} w_{ji} x_j + \varepsilon_i$, where $\varepsilon_i \sim \mathcal{N}\left(0, 1\right)$. For fairness, we prune the DAGs returned by **DrBO**, along with ALIAS and CORL as prescribed in their papers, by thresholding the weight matrix obtained via linear regression at 0.3. This is not done for DAGMA and COSMO because their implementations already incorporated the same pruning scheme.

**Dense graphs.** We stress-test our method with complex structures in Figure 1(a). As depicted in the first column, our method is the only approach that can achieve absolute overall performance in all five metrics, surpassing the second- and third-best methods ALIAS and CORL by large margins of more than 20% in each metric, while gradient-based methods DAGMA and COSMO struggle with much worse performance. From the second and third columns, our **DrBO** approach can reach higher BIC scores, and thus lower SHDs, very sharply with the number of evaluations, highlighting

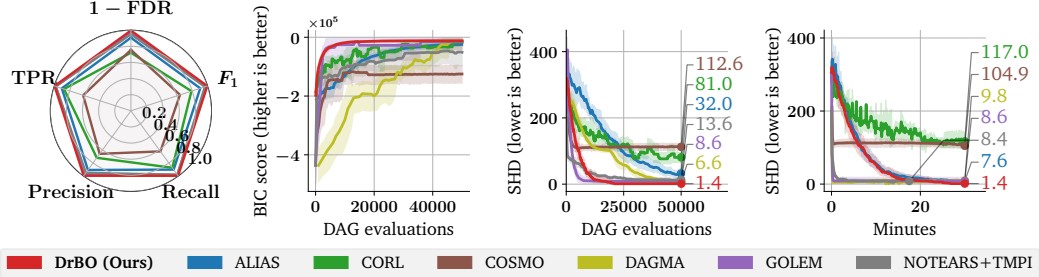

(a) **Dense graphs** with Linear-Gaussian data (DAGs with **30 nodes** and ≈**240 edges**). For fairness, summary metrics (first column) are calculated at 50,000 evaluations for all methods.

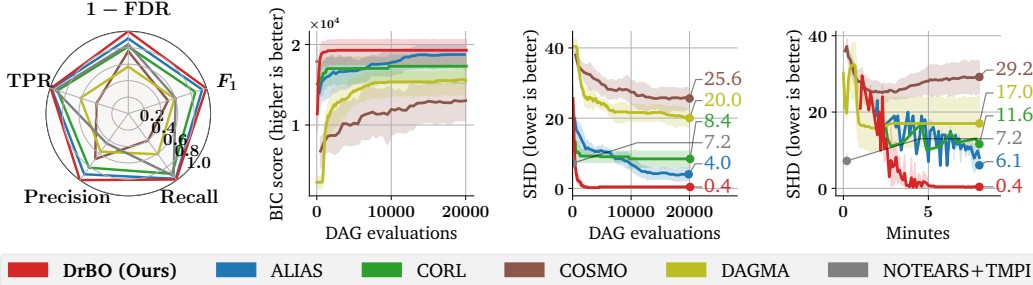

(b) **Large graphs** with Linear-Gaussian data (DAGs with **100 nodes** and ≈**200 edges**). For fairness, summary metrics (first column) are calculated at 50,000 evaluations for all methods.

(c) **Nonlinear data** with Gaussian processes (DAGs with **10 nodes** and ≈**40 edges**). For fairness, summary metrics (first column) are calculated at 20,000 evaluations for all methods.

Figure 1: **DAG learning results on Synthetic data.** *First column:* overall performance in terms of True Positive Rate (TPR, higher is better), Precision, Recall, and $F_1$ score (higher is better), as well as False Discovery Rate (FDR, lower is better). *Second column:* we track the best Bayesian Information Criterion (BIC, higher is better) so far at every optimization step. *Third and Fourth columns:* we monitor the Structural Hamming Distance (SHD, lower is better) of the DAG whose best BIC so far at every optimization step. Shaded areas in the line plots indicate 95% confidence intervals over 5 random datasets. NOTEARS+TMPI usually stops early before the time limit.

its sample-efficiency. Lastly, the fourth column shows that our method's SHD improvement over time is continuous and faster in terms of runtime than all other baselines.

**Large-scale graphs.** Next, we demonstrate the scalability of our method on high-dimensional graphs of 100 nodes, having up to 10,000 edges. Our results in Figure 1(b) shows that **DrBO** is still the leading method for high-dim data, where it obtains absolute overall performance along with the lowest SHD among all methods in limited runtime.

### 5.1.2 NONLINEAR DATA

Following prior works (Zhu et al., 2020; Wang et al., 2021; Yang et al., 2023a; Duong et al., 2025), we demonstrate our method on nonlinear datasets generated using Gaussian processes from Lachapelle et al. (2020). Specifically, we employ the datasets generated according to an ANM $x_i := f_i\left(\mathbf{x}_{\mathrm{pa}_i}\right) + \varepsilon_i$ where $f_i$ is drawn from a Gaussian process prior with a unit bandwidth RBF

kernel, and $\varepsilon_i \sim \mathcal{N}\left(0, \sigma_i^2\right)$ with non-equal noise variances $\sigma_i^2$ sampled uniformly in $[0.4, 0.8]$. ALIAS, CORL, and **DrBO** optimize $S_{\mathrm{BIC-NV}}$, while COSMO, DAGMA, and NOTEARS+TMPI optimize the least-square objective. The empirical results reported in Figure 1(c) confirms that our **DrBO** method also excels on complex nonlinear data in both accuracy and computational cost. This is evidenced by the leading overall performance (first column), along with a vanishing SHD of only $0.4$ at the end of the learning curve, surpassing other methods by a visible gap both in SHD and convergence speed.

## 5.2 RESULTS ON REAL DATA AND STRUCTURES

Table 1: **Causal Discovery Performance on Real-world Structures (Scutari, 2010)**. The performance is measured in Structural Hamming Distance (SHD, lower is better). The numbers are $\mathrm{mean} \pm \mathrm{std}$ over 5 independent datasets with $1{,}000$ observational samples. For fairness, all methods are limited to $20{,}000$ evaluations.

| Dataset
Method | Alarm
(37 nodes, 46 edges) | Asia
(8 nodes, 8 edges) | Cancer
(5 nodes, 4 edges) | Child
(20 nodes, 25 edges) | Earthquake
(5 nodes, 4 edges) |
|---|---|---|---|---|---|
| ALIAS (Duong et al., 2025) | $26.8 \pm 7.8$ | $0.2 \pm 0.5$ | $0.0 \pm 0.0$ | $2.8 \pm 1.6$ | $0.0 \pm 0.0$ |
| CORL (Wang et al., 2021) | $19.8 \pm 8.6$ | $0.0 \pm 0.0$ | $0.0 \pm 0.0$ | $1.6 \pm 2.3$ | $0.0 \pm 0.0$ |
| COSMO (Massidda et al., 2024) | $26.8 \pm 4.0$ | $4.0 \pm 1.6$ | $2.2 \pm 0.8$ | $11.6 \pm 2.3$ | $2.2 \pm 0.8$ |
| DAGMA (Bello et al., 2022) | $25.0 \pm 5.3$ | $2.6 \pm 1.3$ | $1.0 \pm 1.2$ | $8.0 \pm 5.2$ | $1.0 \pm 1.2$ |
| GOLEM (Ng et al., 2020) | $4.8 \pm 6.2$ | $0.0 \pm 0.0$ | $0.0 \pm 0.0$ | $0.0 \pm 0.0$ | $0.0 \pm 0.0$ |
| NOTEARS+TMPI (Zheng et al., 2018; Zhang et al., 2022) | $7.0 \pm 7.9$ | $0.4 \pm 0.9$ | $0.0 \pm 0.0$ | $0.0 \pm 0.0$ | $0.0 \pm 0.0$ |
| **DrBO** (Ours) | $\mathbf{1.0 \pm 2.2}$ | $\mathbf{0.0 \pm 0.0}$ | $\mathbf{0.0 \pm 0.0}$ | $\mathbf{0.0 \pm 0.0}$ | $\mathbf{0.0 \pm 0.0}$ |

**Benchmark data.** We verify the performance of our method on real data using the popular benchmark flow cytometry dataset (Sachs et al., 2005), concerning a protein signaling network based on expression levels of proteins and phospholipids. We employ the observational portion of the dataset containing 853 observations and a known causal network with 11 nodes and 17 edges. We apply the similar settings as in Sec. 5.1.2 for all methods. The evaluations shown in Figure 2 verify the effectiveness of our method on real data, where it effortlessly achieves a lowest SHD of 9 with fewer evaluations compared with the competitors.

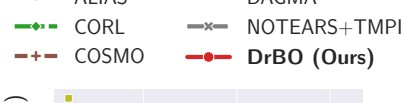

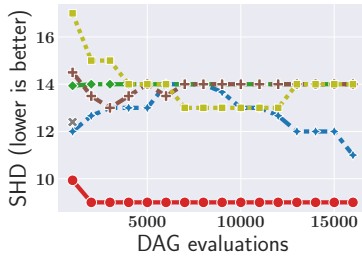

**Real-world structures.** To further illustrate the capabilities of our approach on real-world scenarios, we conduct experiments on real structures provided by the BnLearn repository (Scutari, 2010). Each dataset contains $1{,}000$ observational samples and a ground truth causal network belonging to real-world applications with varying size.

Figure 2: **Causal Discovery Performance on the Benchmark Sachs Dataset (Sachs et al., 2005)**. NOTEARS+TMPI stops early before the max no. of evaluations is reached.

Additional details regarding these datasets are given in Appendix D.3. The results are presented in Table 1, highlighting that our method is the only approach that can consistently achieve zero SHD on four out of five real-world structures. On the Alarm dataset, which appears to be most challenging, our **DrBO** method still leads with much lower SHD compared with all other baselines.

## 5.3 SUPPLEMENTARY RESULTS

**Extended Causal Discovery Settings.** We investigate the performance of **DrBO** in extended scenarios in Appendix F as follows. **Varying sample sizes:** we show in Figure 4 that our method can achieve low SHDs even with limited data. **Different graph models:** in Table 3, **DrBO** achieves low SHDs and surpasses competitors for both ER and SF graphs, even on the dense graphs. **Different noise distributions:** Table 4 shows that **DrBO** also outperforms the baselines under five different noise types. **BGe score:** Figure 5 confirms that our method can work with a different score and match the score of ground truth graphs with low structural errors. **Discrete Data:** in Figure 6, we show that **DrBO** also obtains the highest scores and lowest SHDs compared with the baselines for non-continuous data. **Standardized Data:** we show in Figure 7 that our method is also robust against data standardization for both linear and nonlinear data. **Large-scale Nonlinear Data:** Figure 8 demonstrates **DrBO**'s competitive performance and efficiency for nonlinear data on 50- and 100-node graphs.

**Additional baselines.** In Appendix G, we also examine other baselines that are not based on sequential optimization, showing that **DrBO** also significantly outperforms these methods on linear, nonlinear, as well as real data.

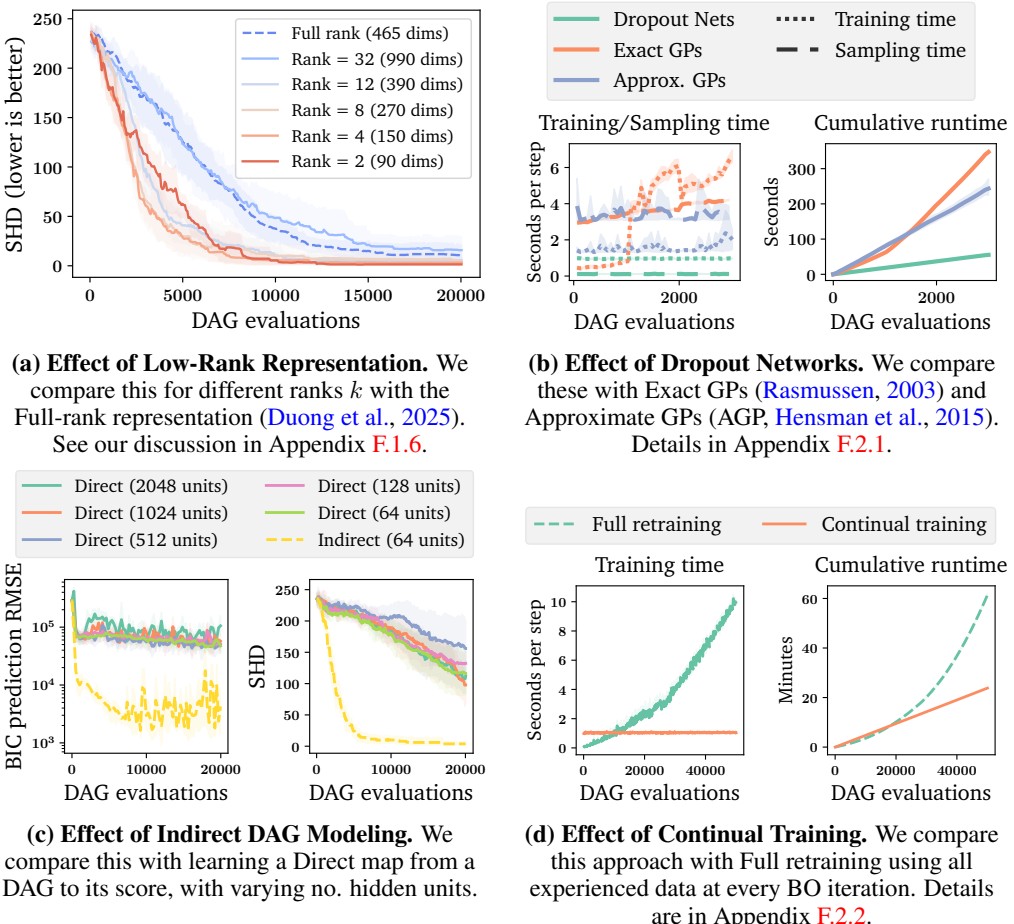

Figure 3: **Ablating our design choices.** All configurations are evaluated on 5 linear-Gaussian datasets of 1,000 samples on 30ER8 graphs. Shaded areas indicate 95% confidence intervals.

**Runtime.** In Appendix H, we also detail the numerical runtime among all methods.

### 5.4 ABLATIONS

In Figure 3, through ablation studies, we justify the design choices outlined in the Introduction, showing that every component contributes considerably to the accuracy and/or scalability of our method. The remaining hyperparameters of our algorithm are also studied in Appendix F.2.

## 6 CONCLUDING REMARKS

This study presents **DrBO**, a novel BO method to search for high-scoring DAGs. We have shown that, by meticulously choosing promising DAGs to evaluate, we can find the optimal one more efficiently and cost-effectively. Our comprehensive experiments demonstrate that **DrBO** performs well even in many intricate settings like dense graphs, high-dimensional, and nonlinear data.

Regarding limitations, in our method, the surrogate model architecture is manually chosen and remains fixed through the course of optimization, and thus is prone underfitting at the end. To mitigate this, incremental neural architecture search techniques (Liu et al., 2018; Geifman & El-Yaniv, 2019) can be employed to facilitate autonomous architecture selection and scaling. In addition, our continual learning component is simple and can benefit from more advanced techniques (Wang et al., 2024) to further improve the performance of our surrogate models.

For future developments, it would be an interesting direction to combine our approach with active causal discovery methods to recover the causal structure even more efficiently. Moreover, our method can be extended to solve causal discovery problems with hidden confounders, where the outputs are no longer DAGs.

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

## A  PROOFS

### A.1  PROOF OF LEMMA 1

*Proof.* The acyclicity of Eq. (4) is ensured by the first term: $H\left(\mathrm{grad}\left(\mathbf{p}\right)\right)$. This is a binary matrix representing the adjacency matrix of a directed graph. In this graph, the presence of an edge $i \to j$ is equivalent to $p_i < p_j$. As such, a cycle $i_1 \to \ldots \to i_1$, if exists, would lead to $p_i < p_i$, which is contradictory, meaning this graph must be a DAG. Multiplying this adjacency matrix with the second term $H\left(\mathbf{R}\cdot\mathbf{R}^\top\right)$, which is also a binary matrix, has the effect of removing already existing edges in this graph, and thus cannot introduce any cycle. This concludes our proof. $\qquad\square$

### A.2  PROOF OF LEMMA 2

*Proof.* For any $\alpha > 0$, we have $H\left(p_i - p_j\right) = H\left(\alpha\left(p_i - p_j\right)\right) = H\left(\alpha p_i - \alpha p_j\right)$ and $H\left(\mathbf{R}\cdot\mathbf{R}^\top\right) = H\left(\alpha^2\mathbf{R}\cdot\mathbf{R}^\top\right) = H\left(\left(\alpha\mathbf{R}\right)\cdot\left(\alpha\mathbf{R}\right)^\top\right)$. Thus, $H\left(\mathrm{grad}\left(\mathbf{p}\right)\right) \odot H\left(\mathbf{R}\cdot\mathbf{R}^\top\right) = H\left(\mathrm{grad}\left(\alpha\mathbf{p}\right)\right) \odot H\left(\left(\alpha\mathbf{R}\right)\cdot\left(\alpha\mathbf{R}\right)^\top\right)$, concluding our proof. $\qquad\square$

## B  DERIVING BIC SCORES

Recall that for a causal model with parameters $\theta := \left\{\{f_i\}_{i=1}^d, P\left(\varepsilon\right)\right\}$, the general BIC is given by

$$S_{\mathrm{BIC}}\left(\mathcal{D}, \mathcal{G}\right) := 2\ln p\left(\mathcal{D} \mid \hat{\theta}, \mathcal{G}\right) - |\mathcal{G}|\ln n, \tag{6}$$

where $\hat{\theta} := \arg\max_\theta p\left(\mathcal{D} \mid \theta, \mathcal{G}\right)$ is the maximum-likelihood estimator of the causal model parameters, $n$ is the sample size of $\mathcal{D}$, and $|\mathcal{G}|$ denotes the number of edges in $\mathcal{G}$. The first term in Eq. (6) is a log-likelihood objective similar to GOLEM (Ng et al., 2020), GraN-DAG (Lachapelle et al., 2020), etc., while the second term penalizes extra edges.

The flexibility of Eq. (6) allows for the adoption of BIC in various causal models by simply specifying the likelihood model. In the following, we derive the BIC scores for additive noise models with non-equal and equal variances, as well as logistic models.

### B.1  BIC FOR ANM WITH NON-QUAL VARIANCES

Let us consider an ANM defined by

$$x_i := f_i\left(\mathbf{x}_{\mathrm{pa}_i^{\mathcal{G}}}\right) + \varepsilon_i, \; \forall i = 1, \ldots, d, \tag{7}$$

where the noise is assumed to be Gaussian with fixed variance: $\varepsilon_i \sim \mathcal{N}\left(0, \sigma_i^2\right) \; \forall i = 1, \ldots d$. The likelihood of a dataset $\mathcal{D} = \left\{\mathbf{x}^{(j)}\right\}_{j=1}^n$ under this model is given by

$$\mathcal{L} = \ln p\left(\mathcal{D} \mid \mathcal{G}, \{f_i\}_{i=1}^d, \{\sigma_i\}_{i=1}^d\right) = -\frac{1}{2}\sum_{i=1}^d \frac{\sum_{j=1}^n \left(x_i^{(j)} - f_i\left(\mathbf{x}_{\mathrm{pa}_i^{\mathcal{G}}}^{(j)}\right)\right)^2}{\sigma_i^2} - \frac{n}{2}\sum_{i=1}^d \ln\sigma_i^2 + \mathrm{constant}. \tag{8}$$

Taking its derivative and setting it to zero, we can solve for the parameters as follows:

$$\hat{\sigma}_i^2 = \underbrace{\frac{1}{n}\sum_{j=1}^n \left(x_i^{(j)} - \hat{f}_i\left(\mathbf{x}_{\mathrm{pa}_i^{\mathcal{G}}}^{(j)}\right)\right)^2}_{\mathrm{MSE}_i}, \tag{9}$$

where $\left\{\hat{f}_i\right\}_{i=1}^d$ are least-square estimators obtained by minimizing $\left(x_i^{(j)} - f_i\left(\mathbf{x}_{\mathrm{pa}_i^{\mathcal{G}}}^{(j)}\right)\right)^2$ over a restricted hypothesis class $f \sim \mathcal{F}$. Following past studies (Zhu et al., 2020; Wang et al., 2021; Yang

et al., 2023a;b; Duong et al., 2025), we use linear regression for linear data and Gaussian process regression for nonlinear data. However, any other regression method can be used as desired.

Subsequently, the first term in Eq. (8) cancels out and the maximum log-likelihood is reduced to

$$\hat{\mathcal{L}} = -\frac{n}{2}\sum_{i=1}^{d}\ln \text{MSE}_i + \text{constant}. \tag{10}$$

Finally, the BIC score for ANMs with non-equal variances is obtained by:

$$S_{\text{BIC-NV}}\left(\mathcal{D}, \mathcal{G}\right) = -n\sum_{i=1}^{d}\ln \text{MSE}_i - |\mathcal{G}|\ln n. \tag{11}$$

### B.2  BIC FOR ANM WITH EQUAL VARIANCES

By setting $\sigma_i = \sigma \ \forall i = 1, \ldots, d$, we repeat the previous steps and obtain the following solution for the maximum likelihood estimators:

$$\hat{\sigma}^2 = \frac{1}{d}\sum_{i=1}^{d}\underbrace{\frac{1}{n}\sum_{j=1}^{n}\left(x_i^{(j)} - \hat{f}_i\left(\mathbf{x}_{\text{pa}_i}^{(j)}\right)\right)^2}_{\text{MSE}_i}. \tag{12}$$

The maximum likelihood now is given by

$$\hat{\mathcal{L}} = -\frac{nd}{2}\ln\frac{\sum_{i=1}^{d}\text{MSE}_i}{d} + \text{constant}. \tag{13}$$

Finally, we obtain the BIC score for ANMs with equal variances as follows:

$$S_{\text{BIC-EV}}\left(\mathcal{D}, \mathcal{G}\right) = -nd\ln\frac{\sum_{i=1}^{d}\text{MSE}_i}{d} - |\mathcal{G}|\ln n. \tag{14}$$

### B.3  BIC FOR BINARY DATA WITH LOGISTIC REGRESSION

From the formulation in Eq. (6), we can also adapt it to non-continuous data. For example, let us consider the logistic causal model governed by

$$x_i \sim \text{Bernoulli}\left(f_i\left(\mathbf{x}_{\text{pa}_i^{\mathcal{G}}}\right)\right). \tag{15}$$

where $f_i$ models the conditional probability of $x_i$ given $\mathbf{x}_{\text{pa}_i^{\mathcal{G}}}$.

The log-likelihood of data under this model is determined by

$$\mathcal{L} = \ln p\left(\mathcal{D} \mid \mathcal{G}, \{f_i\}_{i=1}^{d}\right) = \sum_{i=1}^{d}\sum_{j=1}^{n}\left(x_i^{(j)}\ln f_i\left(\mathbf{x}_{\text{pa}_i^{\mathcal{G}}}^{(j)}\right) + \left(1 - x_i^{(j)}\right)\ln\left(1 - f_i\left(\mathbf{x}_{\text{pa}_i^{\mathcal{G}}}^{(j)}\right)\right)\right).$$

Maximizing this objective and plugging it into Eq. (6) gives us with the BIC for logistic model:

$$S_{\text{BIC-Logistic}}\left(\mathcal{D}, \mathcal{G}\right) = 2\hat{\mathcal{L}} - |\mathcal{G}|\ln n. \tag{16}$$

The first term is equivalent to the logistic loss used in, e.g., DAGMA (Bello et al., 2022), while the second term punishes redundant edges as always.

## C    ADDITIONAL DISCUSSIONS AND ALGORITHM DETAILS

### C.1    PRELIMINARY CANDIDATE GENERATION

In Sec. 4.2, we generate a set of $C$ preliminary candidates in a hyperrectangle centered at the best solution so far $\mathbf{z}^*$. This hyperrectangle can be seen as a single "trust region" in BO (Eriksson et al., 2019; Daulton et al., 2022). More specifically, our trust region is the intersection of the search space $[-1, 1]^{d(1+k)}$ and the hypercube of length $L$ centered at $\mathbf{z}^*$. Following Eriksson et al. (2019), we adaptively update $L$ according to the learning progress. This is done by maintaining a success (and failure) counter that keeps track the number of consecutive BO iterations that improves (or fails to improve, resp.) the DAG score. After $n_{\text{succ}}$ consecutive successes, we enlarge $L$ by two times in order to shift the focus to other regions, and after $n_{\text{fail}}$ consecutive failures, we shrink it by two times to zoom more into the current region. In all experiments, we use the fixed values of $n_{\text{succ}} = 3$, $n_{\text{fail}} = 5$, and $L$ is initialized with the value of $1$ and is clipped to be within $[0.01, 2]$ after each update.

To produce the preliminary candidates on which we generate Thompson samples, for the very first suggestions, following Eriksson et al. (2019), we employ the Latin hypercube design (LHD, McKay et al., 2000), which is a space-filling method used to generate near-random samples from a multi-dimensional space, which ensures that each dimension of the hypercube is evenly covered, while random sampling can lead to an unevenly covered space. For subsequent suggestions, we follow the established procedure in (Eriksson et al., 2019) to first generate a scrambled Sobol sequence (Owen, 1998) within the current trust region, then we use the perturbation value in the Sobol sequence with probability $\min\left\{1, \frac{20}{d(1+k)}\right\}$ for each given candidate and dimension, and the value of the center $\mathbf{z}^*$ otherwise. As noted in (Regis & Shoemaker, 2013; Eriksson et al., 2019), perturbing only a few dimensions can lead to a significant performance improvement for high-dim scenarios.

### C.2    SCALABLE SURROGATE MODELING

The literature of BO is vast and here we only discuss a few promising alternative approaches to scale up surrogate models in BO, and justify of our dropout neural network choice.

**Bayesian neural networks (BNN)** are a also natural replacement thanks to the flexibility of neural networks combined with the inherent ability to model uncertainty of the Bayesian ideology (Springenberg et al., 2016). However, to stay as close as possible to a truly Bayesian treatment, i.e., characterizing the exact posterior distributions, they require stochastic gradient Markov Chain Monte Carlo (MCMC) to sample from the posterior, which necessitates several sampling steps to reach the desirable posterior (Springenberg et al., 2016). Meanwhile, in our method, we sacrifice the accuracy of the posterior inference, so that we can sample from the posterior faster with a single forward pass through the dropout neural networks. That being said, our DAG learning accuracy is still strong, which justifies this sacrifice.

**Sparse GPs (or Approximate GPs)** have also been studied to enhance scalability of GPs (Snelson & Ghahramani, 2005; Hensman et al., 2015; Titsias, 2009) by reducing the number of data points to a set of "inducing points" that are representatives of the true data, which can be chosen to be much smaller than the original data set. However, it is observed that these methods usually do not scale well with dimensionality (Wang et al., 2018).

**Random forest (RF)** has also been considered as an alternative surrogate model in BO (Hutter et al., 2011). An RF is composed of multiple decision trees, each of which is constructed from a portion of the training set and a few random dimensions. Therefore, uncertainty modeling in RF is achieved via the variation of the individual trees' predictions. While RF is known to be efficient, especially in tabular data, we find it more straightforward to train neural networks continually (Wang et al., 2024) compared with tree-based models (Utgoff, 1989; Utgoff et al., 1997). This is crucial for our BO framework to scale with the number of iterations as aforementioned.

**Ensembling methods** (Wang et al., 2018; Guo et al., 2018) sidestep the scalability challenge of BO with the use of an ensemble of models trained on different partitions of the samples and dimensions. A similar idea is also proposed in (Eriksson et al., 2019), where local GPs are trained on multiple local trust regions of the search space. However, these methods require training multiple models,

and managing them is less straightforward than maintaining a single neural network like our method, which also scales very well with a strong causal discovery performance.

### C.3    PRUNING TECHNIQUES

Pruning is typically employed in causal discovery to reduce false positive estimates, and there are several approaches depending on the causal model as follows.

**Linear data.** For linear data, given the resultant DAG $\mathcal{G}$ that needs to be pruned, linear regression is used to find the linear coefficients $\{\hat{w}_{ij}\}_{(i \to j) \in \mathcal{G}}$ associating with the edges in $\mathcal{G}$. Then, only the weights satisfying a certain absolute strength $\alpha$ is kept, while the remaining are removed. In other words, the edge $(i \to j) \in \mathcal{G}$ is kept iff $|\hat{w}_{ij}| > \alpha$. Typically, the weights in the true generative process for linear models are sampled from $\mathcal{U}([-2, -0.5] \cup [0.5, 2])$, and the common value for $\alpha$ is 0.3 (Zheng et al., 2018; Zhu et al., 2020; Wang et al., 2021; Bello et al., 2022; Massidda et al., 2024; Duong et al., 2025).

**Nonlinear additive model.** For nonlinear data under additive models, a common approach is based on feature selection using generalized additive model (GAM) regression (Bühlmann et al., 2014), also known as CAM pruning. Particularly, each node $i$ is regressed on its parents $\mathbf{x}_{\text{pa}_i^{\mathcal{G}}}$ using GAM, then the significance test of covariates is conducted, and a parent is kept if its $p$-value is lower than the significance level of 0.001. This is also done in (Zhu et al., 2020; Wang et al., 2021; Massidda et al., 2024; Rolland et al., 2022; Sanchez et al., 2023; Duong et al., 2025).

**Non-additive models.** For general non-additive models, conditional independence (CI) testing can also be employed to prune edges. To be more specific, the Faithfulness assumption (Peters et al., 2017) implies that any conditional independence observed in data reflects the corresponding $d$-separation in the causal graph, so if $x_i \perp\!\!\!\perp x_j \mid \mathbf{x}_{\text{pa}_i \setminus \{j\}}$ for some $j \in \text{pa}_i$, then $j$ is an extra edge and needs to be removed. This follows the same idea of constraint-based causal discovery (Spirtes et al., 2000; Colombo et al., 2012) and feature selection via Markov Blankets (Koller & Sahami, 1996; Xing et al., 2001). As observed in (Duong et al., 2025), CI-based pruning leads to better performance on the Sachs dataset compared with CAM pruning, and hence we also employ it for all methods on the Sachs dataset. The specific CI test is the popular kernel-based method KCIT (Zhang et al., 2011) with a significance level of 0.001.

## D    EXPERIMENT DETAILS

### D.1    DAG LEARNING METRICS

We evaluate the performance of each method using the following standard measures, each of which calculates the disparity between an estimated DAG and the ground truth DAG:

- **Structural Hamming Distance (SHD, lower is better):** this is the most common metric in causal discovery, which counts the minimum number of edge additions, removals, and reversals to turn the estimated DAG into the true graph.

- **BIC score (higher is better):** we also monitor the BIC score of the estimated DAG for all methods in the main text (despite the fact that some baselines do not optimize this score). For linear data, we use the BIC with equal variances and linear regression, while for nonlinear data, BIC with non-equal variance with Gaussian process regression is used, and BIC with logistic regression is employed for binary data.

- **True Positive Rate (TPR, higher is better):** this measures the ratio of correctly recovered edges over the true edges in the ground truth DAG.

- **False Discovery Rate (FDR, lower is better):** this measures the proportion of incorrectly estimated edges over all estimated edges.

- **Precision, Recall, and** $F_1$: this measures the binary classification performance by treating the binary adjacency matrix as a set of individual binary classification tasks.

## D.2 SYNTHETIC CAUSAL GRAPHS

Our synthetic causal graphs involve one of the two well-known graph models:

- **Erdős-Rényi** (ER, Erdős & Rényi, 1960): the edges in this type of graph are added independently with fixed probability. To generate a DAG of $d$ nodes with an expected in-degree of $e$, we first generate an undirected graph where edges are added with probability $\frac{4de}{d(d-1)}$, then orient the edges using a random permutation over the list of nodes.

- **Scale-Free** (SF, Barabási & Albert, 1999): these are graphs where the degree distribution follows a power law, where a few nodes have many connections while others have only a few connections. To generate SF graphs with $\approx de$ edges, we start with an empty graph then repeatedly grow it by attaching new nodes, each with $k$ edges, that are preferentially attached to existing nodes.

*Remark* 1. It is noteworthy that the majority of methods perform well only for graphs with up to $e = 4$ (Zheng et al., 2018; Zhu et al., 2020; Ng et al., 2020; Yu et al., 2021; Wang et al., 2021; Bello et al., 2022; Yang et al., 2023b). Indeed, it was noted in (Bello et al., 2022) that ER4 and SF4 are the hardest settings. However, in this study, we have shown that our **DrBO** method is still very accurate on much denser ER8 and SF8 graphs.

## D.3 BNLEARN DATASETS

The BnLearn repository[2] (Scutari, 2010) contains a set of Bayesian networks of varying sizes and complexities from different real-world domains. The chosen networks in our study include Alarm (Beinlich et al., 1989), Asia (Lauritzen & Spiegelhalter, 1988), Cancer (Korb & Nicholson, 2010), Child (Spiegelhalter, 1992), and Earthquake (Korb & Nicholson, 2010). However, these datasets only describe the conditional probability distributions of discrete variables, while the baselines considered in our method are mostly implemented for continuous data. Since the purpose of this experiment is to show that *our method can correctly recover real structures*, we use the networks from the BnLearn to generate continuous data. Specifically, we employ linear-Gaussian SCMs as in Sec. 5.1.1 to generate synthetic data adhering to real causal networks, and each dataset contains only 1,000 observational samples.

## D.4 IMPLEMENTATIONS AND PLATFORM

**Implementations.** We employ the following implementations for the considered baselines as follows:

- DAGMA (Bello et al., 2022): we use the official implementation provided by the authors at `https://github.com/kevinsbello/dagma`.
- COSMO (Massidda et al., 2024): we use the original implementation attached as supplementary material at `https://openreview.net/forum?id=KWO8LSUC5W`.
- CORL (Wang et al., 2021): we employ the official implementation provided in the gCastle library (Zhang et al., 2021) at `https://github.com/huawei-noah/trustworthyAI`.
- ALIAS (Duong et al., 2025): we reimplement their method by following the exact instructions and libraries described.
- GOLEM (Ng et al., 2020): we adopt the implementation provided by gCastle (Zhang et al., 2021) at `https://github.com/huawei-noah/trustworthyAI`.
- NOTEARS+TMPI (Zheng et al., 2018; Zhang et al., 2022): we replace the DAG constraint in NOTEARS' implementation at `https://github.com/xunzheng/notears` with the TMPI constraint at `https://github.com/zzhang1987/Truncated-Matrix-Power-Iteration-for-Differentiable-DAG-Learning`.

**Platform.** The majority of our experiments are conducted on a machine with Intel® Core™ i9-13900KF processor and NVIDIA RTX 4070 Ti GPU. The only exception is the case of CORL on

---

[2]Data is publicly downloadable at `https://www.bnlearn.com/bnrepository`

large graphs (Figure 1(b)) which requires more than 16Gb of CUDA memory, and therefore these experiments are conducted on an NVIDIA A100 GPU with 40G of CUDA memory instead.

Additionally, in this study, we evaluate the performance of various methods with respect to both the number of steps and runtime, addressing two independent questions: *"How accurate can a method become given a fixed number of steps?"* and *"How accurate can it be within a given runtime?"*. To ensure fair comparisons, we account for potential biases in measuring performance solely by the number of DAG evaluations. This is particularly important for methods like gradient-based approaches (e.g., DAGMA), which may require many steps but still exhibit low overall runtime.

To address this, we use runtime as a more equitable efficiency metric. Specifically, we set a high number of steps for all methods (e.g., we use $T = 800$ iterations instead the default of only $T = 5$ for DAGMA on linear data) and disable early stopping if applicable, to capture their progression over an extended period of time. We then truncate the tracking data, which contains performance metrics and timestamp at every step, either at a fixed number of steps or a specified runtime, as illustrated in Figure 1. This ensures that the results in the last column of Figure 1 are not constrained by the number of steps. For instance, at the 5-minute mark in the last column of Figure 1a, DAGMA completes approximately 5 million steps compared to only 50,000 steps in the third column.

## E  HYPERPARAMETERS

We provide the specific set of hyperparameters for our **DrBO** method in Table 2. More details can be found in our published source code.

Table 2: Hyperparameters for **DrBO**. Unless specifically indicated, the default hyperparameters here are used for all experiments. More details can be found in our published source code. [†]For nonlinear data with GPs, we use GP regression with regularization $\alpha = 1$ (because the additive noises have near-unit variances) and radial basis function (RBF) kernel, where the length scale is optimized over $(1, 10^5)$. [‡]For the Sachs dataset, we also use GP regression with the same kernel as above. In addition, as the noise variances are unknown, we set a really small value $\alpha = 10^{-8}$ just to ensure positive definiteness of the covariance matrix, and following Wang et al. (2021); Duong et al. (2025), we also employ the median bandwidth heuristic for the kernel, by dividing the predictors by their median pairwise euclidean distance before GP regression is applied.

| Hyperparameter | Experiment | | |
| --- | --- | --- | --- |
| | **Linear data** | **Nonlinear data with GPs** | **Sachs data** |
| Normalize data | No | No | Yes |
| Scoring function | $S_{\text{BIC-EV}}$ with linear regression | $S_{\text{BIC-NV}}$ with GP regression[†] | $S_{\text{BIC-NV}}$ with GP regression[‡] |
| Pruning method | Linear pruning | No pruning | CIT pruning |
| Batch size $B$ | | 64 | |
| DAG rank $k$ | | 8 | |
| No. training steps $n_{\text{grads}}$ | | 10 | |
| No. preliminary candidates $C$ | | 100,000 | |
| Optimizer | | Adam | |
| Learning rate | | 0.1 | |
| Replay buffer size $n_{\text{replay}}$ | | 1,024 | |
| No. hidden units | | 64 | |
| Dropout rate | | 0.1 | |

For ALIAS and CORL, apart from the recommended default hyperparameters, for fairness, we also use the same batch size of 64 as ours (except for CORL which requires the batch size to be at least the number of nodes, so for 100-node graphs we have to set the batch size to 100), as well as the same scoring function and pruning method as ours in each experiment. Regarding COSMO and DAGMA, we use the linear and nonlinear versions specific to each experiment and the hyperparameters are set as recommended. We also use the same BIC score and pruner specific to each experiment to track the progress of all methods in Figure 1.

Regarding the number of evaluations, for all methods, we run more than needed then cut off at common thresholds, as mentioned in Appendix D.4.

*Remark* 2. Following the same line as in (Bello et al., 2022; Zheng et al., 2018) and many studies in this field, we intentionally avoid hyperparameter tuning. This is to prevent injecting spurious

information of the dataset characteristics into the causal discovery results, which is against the idea of causality. Therefore, we use a fixed set of hyperparameters for each method in each setting. The hyperparameters of baseline methods are chosen as recommended in the original manuscripts, while ours are chosen based on common practice and prior experience. Indeed, our ablation studies reveal that there are better hyperparameter choices that can further improve the performance of our method.

# F    ADDITIONAL CAUSAL DISCOVERY SETTINGS

## F.1    DIFFERENT SAMPLE SIZES

In Figure 4, we test our method with varying sample sizes on linear-Gaussian data. The results indicate that our method is already very accurate with SHD $\approx 0$ at merely $500$ samples.

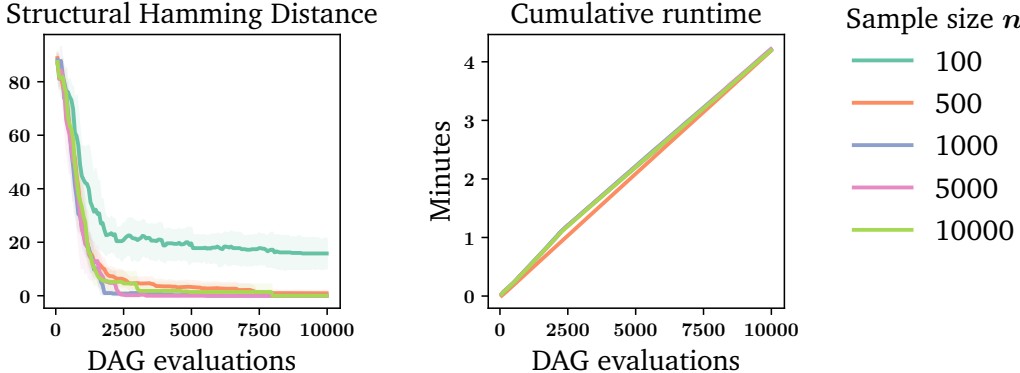

Figure 4: **Causal Discovery Performance with Varying Sample Sizes.** We apply our **DrBO** method on linear-Gaussian data with 20ER4 graphs. Shaded areas represent 95% confidence interval over 5 runs.

### F.1.1    DIFFERENT GRAPH TYPES

We evaluate our method on different graph types, namely ER and SF, with varying densities, as shown in Table 3. Our method perfectly identifies the correct DAG in all cases for both graph models, even in the dense graphs ER8 and SF8.

Table 3: **Causal Discovery Performance with Different Graph Types.** The considered graph models include Erdős-Rényi (Erdős & Rényi, 1960) and Scale-Free (SF, Barabási & Albert, 1999), with expected in-degrees of 2 and 8 corresponding to the case of sparse and dense structures, respectively. We compare our **DrBO** method with ALIAS (Duong et al., 2025) and DAGMA (Bello et al., 2022) on linear-Gaussian datasets with 20 variables and 10,000 samples. The numbers are mean $\pm$ std over 5 random datasets. For fairness, all methods are limited to 10,000 score evaluations.

| Graph | Expected In-degree | Method | SHD $\downarrow$ | FDR $\downarrow$ | TPR $\uparrow$ |
|---|---|---|---|---|---|
| ER | 2 | ALIAS | $21.6 \pm 7.2$ | $0.32 \pm 0.12$ | $0.68 \pm 0.08$ |
| | | DAGMA | $25.8 \pm 7.3$ | $0.28 \pm 0.13$ | $0.55 \pm 0.07$ |
| | | **DrBO** (ours) | $\mathbf{0.0 \pm 0.0}$ | $\mathbf{0.00 \pm 0.00}$ | $\mathbf{1.00 \pm 0.00}$ |
| | 8 | ALIAS | $84.2 \pm 7.5$ | $0.15 \pm 0.03$ | $0.53 \pm 0.04$ |
| | | DAGMA | $122.4 \pm 4.7$ | $0.19 \pm 0.03$ | $0.28 \pm 0.02$ |
| | | **DrBO** (ours) | $\mathbf{0.0 \pm 0.0}$ | $\mathbf{0.00 \pm 0.00}$ | $\mathbf{1.00 \pm 0.00}$ |
| SF | 2 | ALIAS | $13.6 \pm 2.7$ | $0.18 \pm 0.09$ | $0.71 \pm 0.02$ |
| | | DAGMA | $23.4 \pm 9.8$ | $0.25 \pm 0.19$ | $0.53 \pm 0.13$ |
| | | **DrBO** (ours) | $\mathbf{0.0 \pm 0.0}$ | $\mathbf{0.00 \pm 0.00}$ | $\mathbf{1.00 \pm 0.00}$ |
| | 8 | ALIAS | $68.8 \pm 9.0$ | $0.34 \pm 0.08$ | $0.52 \pm 0.06$ |
| | | DAGMA | $94.4 \pm 8.1$ | $0.49 \pm 0.11$ | $0.2 \pm 0.06$ |
| | | **DrBO** (ours) | $\mathbf{0.0 \pm 0.0}$ | $\mathbf{0.00 \pm 0.00}$ | $\mathbf{1.00 \pm 0.00}$ |

### F.1.2 DIFFERENT NOISE DISTRIBUTIONS

To examine the performance of our method compared with others under noise misspecification, in Table 4, we evaluate causal discovery performance various noise distributions. Specifically, we consider linear SCM $x_i := \sum_{j \in \mathrm{pa}_i} w_{ji} x_j + \varepsilon_i$, where $\varepsilon_i$ is drawn from one of the following distributions

- **Exponential noise:** $\varepsilon_i \sim \mathrm{Exp}\,(1)\,, \forall i = 1, \ldots, d$, where 1 is the scale parameter.

- **Gaussian noise:** $\varepsilon_i \sim \mathcal{N}\,(0, 1)\,, \forall i = 1, \ldots, d$, where 0 is the mean and 1 is the variance.

- **Gumbel noise:** $\varepsilon_i \sim \mathrm{Gumbel}\,(0, 1)\,, \forall i = 1, \ldots, d$, where 0 is the location parameter and 1 is the scale parameter.

- **Laplace noise:** $\varepsilon_i \sim \mathrm{Laplace}\,(0, 1)\,, \forall i = 1, \ldots, d$, where 0 is the location parameter and 1 is the scale parameter.

- **Uniform noise:** $\varepsilon_i \sim \mathcal{U}\,(-1, 1)\,, \forall i = 1, \ldots, d$, where the minimum value is $-1$ and maximum value is 1.

*Remark* 3. We note that our BIC score is kept unchanged for different noises to show that our method can work well beyond the assumed Gaussian noise.

Table 4: **Causal Discovery Performance with Different Noise Distributions.** We consider 5 noise distributions: Exponential, Gaussian, Gumbel, Laplace, and Uniform. Our **DrBO** method is compared with ALIAS (Duong et al., 2025) and DAGMA (Bello et al., 2022) on linear-Gaussian datasets with 20ER4 graphs and $10,000$ samples. The numbers are mean $\pm$ std over 5 random datasets. For fairness, all methods are limited to $20,000$ score evaluations.

| Noise Type | Method | SHD $\downarrow$ | FDR $\downarrow$ | TPR $\uparrow$ |
|---|---|---|---|---|
| Exponential | ALIAS | $36.2 \pm 13.3$ | $0.24 \pm 0.07$ | $0.75 \pm 0.09$ |
| | DAGMA | $56.6 \pm 6.8$ | $0.22 \pm 0.10$ | $0.37 \pm 0.05$ |
| | **DrBO** (ours) | $\mathbf{0.4 \pm 0.6}$ | $\mathbf{0.00 \pm 0.01}$ | $\mathbf{0.99 \pm 0.01}$ |
| Gaussian | ALIAS | $34.2 \pm 11.4$ | $0.23 \pm 0.08$ | $0.77 \pm 0.05$ |
| | DAGMA | $57.2 \pm 7.6$ | $0.25 \pm 0.09$ | $0.39 \pm 0.09$ |
| | **DrBO** (ours) | $\mathbf{0.0 \pm 0.0}$ | $\mathbf{0.00 \pm 0.00}$ | $\mathbf{1.00 \pm 0.00}$ |
| Gumbel | ALIAS | $36.6 \pm 12.9$ | $0.26 \pm 0.08$ | $0.77 \pm 0.06$ |
| | DAGMA | $54.8 \pm 7.4$ | $0.25 \pm 0.04$ | $0.44 \pm 0.08$ |
| | **DrBO** (ours) | $\mathbf{0.2 \pm 0.5}$ | $\mathbf{0.00 \pm 0.01}$ | $\mathbf{1.00 \pm 0.01}$ |
| Laplace | ALIAS | $36.4 \pm 14.2$ | $0.26 \pm 0.07$ | $0.76 \pm 0.09$ |
| | DAGMA | $56.0 \pm 8.8$ | $0.25 \pm 0.08$ | $0.40 \pm 0.09$ |
| | **DrBO** (ours) | $\mathbf{0.0 \pm 0.0}$ | $\mathbf{0.00 \pm 0.00}$ | $\mathbf{1.00 \pm 0.00}$ |
| Uniform | ALIAS | $34.4 \pm 13.0$ | $0.24 \pm 0.08$ | $0.77 \pm 0.05$ |
| | DAGMA | $59.0 \pm 7.3$ | $0.26 \pm 0.08$ | $0.39 \pm 0.09$ |
| | **DrBO** (ours) | $\mathbf{0.0 \pm 0.0}$ | $\mathbf{0.00 \pm 0.00}$ | $\mathbf{1.00 \pm 0.00}$ |

### F.1.3 BGe Score for Markov Equivalence Class Discovery

Here, we show that our method is not restricted to the BIC score and can be extended to other scores as well. We consider the Bayesian Gaussian equivalent (BGe, Geiger & Heckerman, 1994; Heckerman et al., 1995) score for learning the Markov Equivalence Class (MEC) of DAGs in linear-Gaussian settings. The BGe score assigns equal scores to DAGs belonging to the same MEC, and can be decomposed as the sum of local scores as follows:

$$S_{\text{BGe}}\left(\mathcal{D}, \mathcal{G}\right) = \sum_{i}^{d} \text{LocalBGe}_i \left(\text{pa}_i^{\mathcal{G}}\right). \tag{17}$$

We refer readers to, for example, Kuipers et al. (2014), for the specific formula of the BGe score. To adapt our method with this score, we simply train each dropout network $\text{DropoutNN}_i$ to predict $\text{LocalBGe}_i$ from $\text{pa}_i^{\mathcal{G}}$ and combine them using Eq. (17). It can be seen that this adaptation does not involve changing any other component of our method.

We report the results in Figure 5, where we compare **DrBO** with two popular baselines that are well-known to recover the MEC, namely PC (Spirtes et al., 2000) and GES (Chickering, 2002).[3] The results illustrate that our method can find the DAGs with the highest BGe scores, while the scores from PC' and GES's estimations are well below those of the ground truths. As a result, our recovered structures are more accurate than the baselines.

---

[3]We use the gCastle library (Zhang et al., 2021) for their implementations, where hyperparameters are left as default.

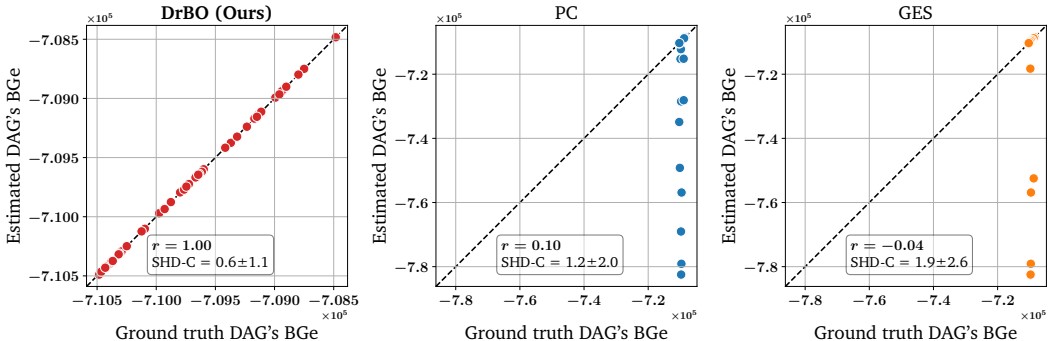

Figure 5: **BGe for Markov Equivalence Class Discovery.** We compare the BGe score of ground truth DAGs and the estimations from **DrBO** with two popular baselines PC (Spirtes et al., 2000) and GES (Chickering, 2002). Each point corresponds to one of 50 random datasets with linear Gaussian data on ER graphs of 5 nodes and 5 edges on average. The Pearson correlation coefficient $r$ between the scores of the estimated and ground truth DAGs are included. In addition, we also report the SHD-C metric, which measures the structural distance between MECs.

### F.1.4 DISCRETE DATA

In this section, we show that our method is not limited to continuous data either. To demonstrate, we consider binary data with logistic causal models:

$$x_i \sim \text{Bernoulli}\left(f_i\left(\mathbf{x}_{\text{pa}_i^{\mathcal{G}}}\right)\right).$$

We adapt our method to this situation by simply changing the BIC score to take into account logistic models. More particularly, we use the BIC score for logistic data as in Eq. (16), where the maximum log-likelihood can be decomposed into

$$\hat{\mathcal{L}} = \sum_{i=1}^{d} \text{LocalMLL}_i,$$

where $\text{LocalMLL}_i = \sum_{j=1}^{n} \left(x_i^{(j)} \ln \hat{f}_i\left(\mathbf{x}_{\text{pa}_i^{\mathcal{G}}}^{(j)}\right) + \left(1 - x_i^{(j)}\right) \ln \left(1 - \hat{f}_i\left(\mathbf{x}_{\text{pa}_i^{\mathcal{G}}}^{(j)}\right)\right)\right)$, with $\hat{f}_i$ being the maximum-likelihood estimator found via logistic regression.

Then, we employ our local surrogate models $\text{DropoutNN}_i$ to model $\text{LocalMLL}_i$, and the acquisition function values are obtained by summing the local score samples. Again, this adaptation only involves changing the scoring function and does not modify any other component of our BO framework.

We report the causal discovery performance under binary data in Figure 6, where we also compare **DrBO** with the state-of-the-art DAGMA, which is now set to use the logistic loss supported. It can be seen that our method also performs well for binary data, where it consistently obtains the highest scoring graphs, resulting in a low structural error. Meanwhile, DAGMA usually finds only sub-optimal solutions which lead to high structural errors.

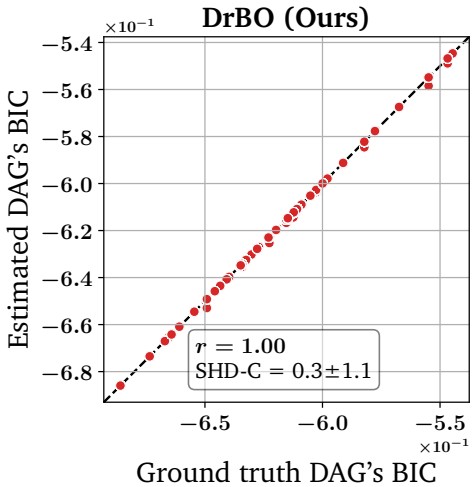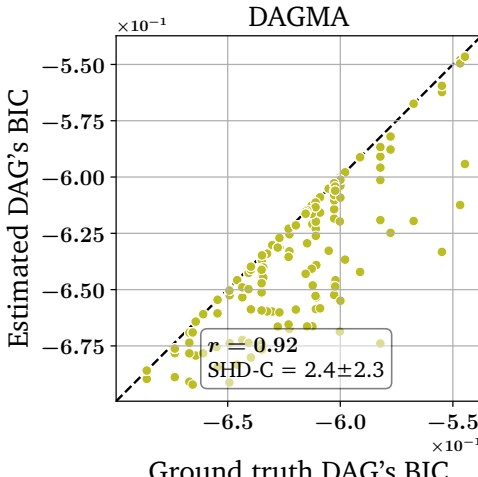

Figure 6: **Causal Discovery performance on Binary Data.** We compare our **DrBO** method using the BIC score for the logistic model with DAGMA (Bello et al., 2022) using the logistic loss. Each point corresponds to one of 50 random datasets with logistic data on ER graphs of 5 nodes and 5 edges on average. The Pearson correlation coefficient $r$ between the scores of the estimated and ground truth DAGs are included. In addition, we also report the SHD-C metric, which measures the structural distance between MECs.

### F.1.5 STANDARDIZED DATA

As discussed in Reisach et al. (2021), marginal variances may contain crucial information about the causal ordering among the causal variables, and thus revealing the causal DAG by simply sorting the variables by increasing variances. For this reason, in this section, we investigate the causal discovery performance of the proposed method in comparison with the baselines under uninformative marginal variances, by standardizing the observed data to have zero mean and unit variance per dimension, before feeding it to causal discovery methods.

**Linear-Gaussian data.** Since the noise variances are non-equal after standardization, we employ the non-equal variance versions of the methods that support it, including ALIAS, CORL, GOLEM, and **DrBO**. In addition, as the data is standardized, the usual threshold of 0.3 for pruning is no longer appropriate because significant edge weights may be rescaled to much smaller values after standardization, so in this experiment, we increase the sample size to 100,000 to reduce weight estimation variance, and lower the pruning threshold to 0.01. The results presented in Figure 7(a) confirm that our method is still robust for standardized data. Overall, while all methods obtain a non-zero SHD due to the difficulty of standardized data, which render the equal-variance linear-Gaussian SCM unidentifiable, our method still outperforms other baselines significantly, where we achieve an SHD≈3, while the second-best SHD is nearly 20, highlighting **DrBO**'s improved effectiveness over existing approaches in this intricate scenario. Additionally, even though the same pruning threshold is used for all methods, our method barely predicts any extra edge, while other baselines still suffer from high false discovery rate. Moreover, our method does not predict too many reverse edges, as opposed to most methods, showing that while data standardization negatively impacts causal discovery performance to some extent, the effect on our method is minimal.

**Nonlinear data with GPs**. Figure 7(b) presents the results for nonlinear data with GPs with standardization, showing that our method can achieve a very low SHD and surpasses other methods considerably. This result is similar to Figure 1(c), where the same datasets employed are not standardized, indicating that the performance of our method is not affected by data standardization in this case, which could be potentially thanks to the fact that nonlinear ANMs remain identifiable after standardization.

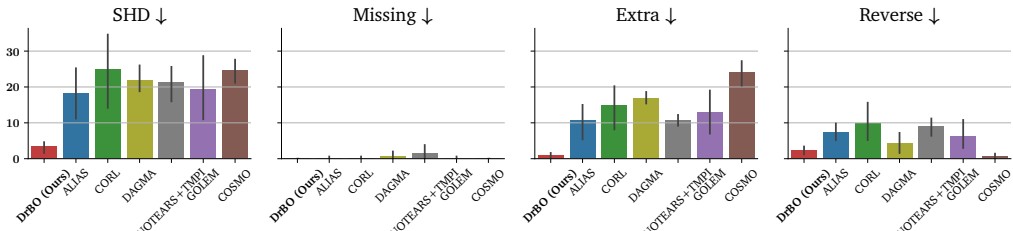

**(a) Linear-Gaussian Data** (DAGs with **10 nodes** and ≈**20 edges**). For fairness, all metrics are calculated at 20,000 evaluations for all methods.

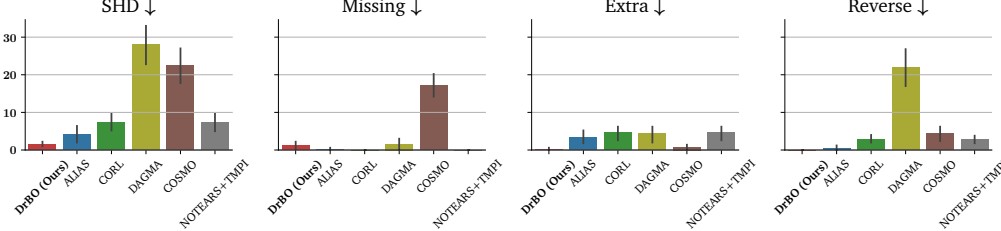

**(b) Nonlinear Data with GPs** (DAGs with **10 nodes** and ≈**40 edges**). For fairness, all metrics are calculated at 20,000 evaluations for all methods.

Figure 7: **Causal Discovery Performance on Standardized Data.** Performance metrics are Structural Hamming Distance (SHD), number of Missing, Extra, and Reverse edges. Lower values are more preferable. Error bars indicate 95% confidence intervals over 5 simulations.

### F.1.6 WHY LOW-RANK DAG REPRESENTATION TENDS TO PERFORM BETTER

In Figure 3(a), we have shown that the rank $k$ of our DAG representation plays an important role in the sample-efficiency of our method, where lower ranks clearly enable reaching the accurate DAGs earlier than their high-rank counterparts. This is because it is much more challenging to search in a very high-dimensional space compared to a lower-dimensional one. Specifically, for higher ranks, the search space is much larger and sparser than the low-rank ones, and due to the curse of dimensionality, sampling the same number of random DAG candidates (step 3 in Algorithm 1) in the higher-rank search spaces tends to lead to fewer unique candidates compared with a lower-rank one, reducing the chance to meet the higher-scoring DAGs earlier.

To empirically verify this, we calculate the number of unique DAGs among 1,000 random 30-node DAGs generated with different ranks in Table 5. It can be seen that, typically, the lower the rank, the more unique DAGs we can pre-examine for exploration. For $k = 2$, almost every DAG among 1,000 generated DAGs is unique, and thus our method can reach SHD≈0 very quickly in Figure 3(a), whereas the full-rank representation is higher-dimensional and can only generate fewer than half the unique DAGs. In addition, for $k = 32 \approx d$, where the dimensionality is highest, we can only generate fewer than 10% of unique DAGs, explaining why this representation is the least sample-efficient one in Figure 3(a).

Table 5: **Effect of DAG Rank on Exploration Diversity.** We generate 1,000 DAGs with $d = 30$ nodes using $\mathcal{G} := \tau(\mathbf{z}), \mathbf{z} \in [-1, 1]^{d \cdot (1+k)}$ with different $k$. The numbers are mean ± std over 10 simulations.

| Rank $k$ in Eq. (4) | Number of dimensions | Number of unique 30-node DAGs over 1,000 random DAGs |
|---|---|---|
| 2 | 90 | $926.7 \pm 7.0$ |
| 4 | 150 | $779.2 \pm 12.7$ |
| 8 | 270 | $493.5 \pm 12.3$ |
| 12 | 390 | $332.4 \pm 10.8$ |
| 32 | 990 | $90.7 \pm 9.5$ |
| Full rank (Vec2DAG, Duong et al., 2025) | 465 | $421.9 \pm 13.8$ |

### F.1.7 Large-scale nonlinear data

For higher-dimensional data with nonlinearity, following Zhang et al. (2022), we evaluate our method on 50ER2 and 100ER1 graphs with nonlinear SCM $\mathbf{x} := \mathbf{B} \cdot \cos{(\mathbf{x})} + \varepsilon$, where the weights $\mathbf{B}$ are sampled uniformly in $[-2, -0.5] \cup [0.5, 2]$ and $\varepsilon_i \sim \mathcal{N}(0, 1)$. This model is identifiable according to Bühlmann et al. (2014). To calculate the BIC score, we employ linear regression with cosine features. The results reported in Figure 8 demonstrate that our method is also competitive on large-scale nonlinear data, where it can outperform the baseline in both causal discovery performance and runtime by a visible margin.

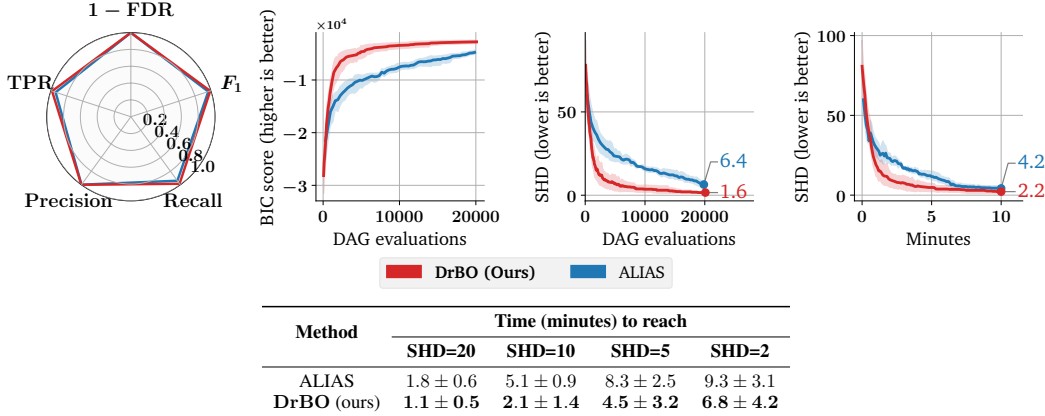

| Method | Time (minutes) to reach | | | |
| --- | --- | --- | --- | --- |
| | SHD=20 | SHD=10 | SHD=5 | SHD=2 |
| ALIAS | $1.8 \pm 0.6$ | $5.1 \pm 0.9$ | $8.3 \pm 2.5$ | $9.3 \pm 3.1$ |
| **DrBO** (ours) | $\mathbf{1.1 \pm 0.5}$ | $\mathbf{2.1 \pm 1.4}$ | $\mathbf{4.5 \pm 3.2}$ | $\mathbf{6.8 \pm 4.2}$ |

**(a) 50 nodes** and ≈**100 edges**. For fairness, summary performance metrics (circular plot) are calculated at 20,000 evaluations for both methods.

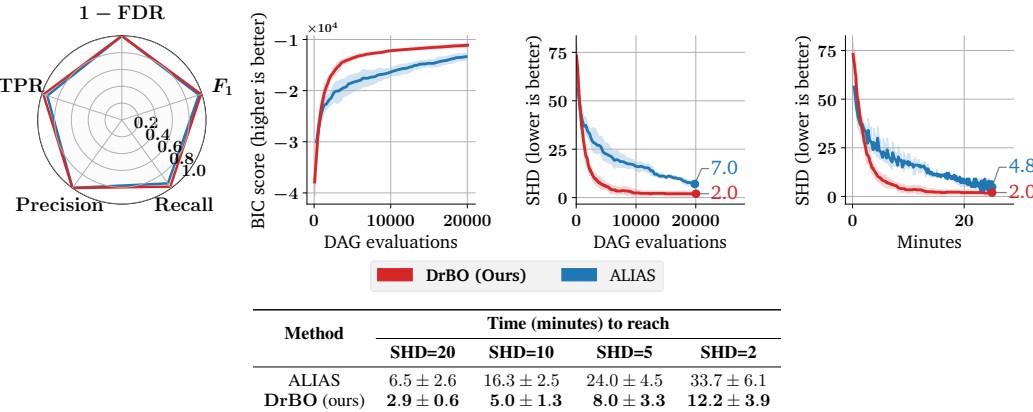

| Method | Time (minutes) to reach | | | |
| --- | --- | --- | --- | --- |
| | SHD=20 | SHD=10 | SHD=5 | SHD=2 |
| ALIAS | $6.5 \pm 2.6$ | $16.3 \pm 2.5$ | $24.0 \pm 4.5$ | $33.7 \pm 6.1$ |
| **DrBO** (ours) | $\mathbf{2.9 \pm 0.6}$ | $\mathbf{5.0 \pm 1.3}$ | $\mathbf{8.0 \pm 3.3}$ | $\mathbf{12.2 \pm 3.9}$ |

**(b) 100 nodes** and ≈**100 edges**. For fairness, summary performance metrics (circular plot) are calculated at 20,000 evaluations for all methods.

Figure 8: **DAG learning results on large-scale nonlinear data.**

## F.2 Ablation Experiments

### F.2.1 Effect of Dropout Networks

In Figure 3(b), we compare our dropout networks as the surrogate model with exact GPs and approximate GPs. Approximate GPs learn a set of pseudo data points called inducing points and conduct inference via these points instead of the real data (Hensman et al., 2015). Here we use a small number of 100 inducing points for Approximate GPs, to see they can scale well with few inducing points. Due to the limited scalability and intensive memory requirement for GPs, we can only use $C = 10,000$ preliminary candidates on which we sample from the posteriors, while our default hyperparameter is $C = 100,000$ using dropout networks.

### F.2.2  EFFECT OF CONTINUAL TRAINING

In Figure 3(d), we compare the continual training approach with fully retraining using all data. For continual learning, we use default hyperparameters $n_{\text{replay}} = 1{,}024$, $B = 64$, and $n_{\text{grads}} = 10$, meaning for each BO iteration, we perform 10 gradients update, each update is calculated from $1{,}024 + 64 = 1{,}088$ datapoints. To ensure fairness, we also use $n_{\text{grads}} = 10$ epochs and a mini-batch size of $1{,}088$ for the full retraining approach.

### F.2.3  EFFECT OF EVALUATION BATCH SIZE ($B$)

In Figure 9, we show the influence of the evaluation batch size $B$ onto the performance and scalability of our method. Overall, it is clear that smaller batch sizes lead to better SHD but much worse runtime since the surrogate model is updated more frequently, and vice versa. However, $B = 64$ seems to achieve balance, where it enables SHD $\approx 0$ and lower runtime than smaller batch sizes.

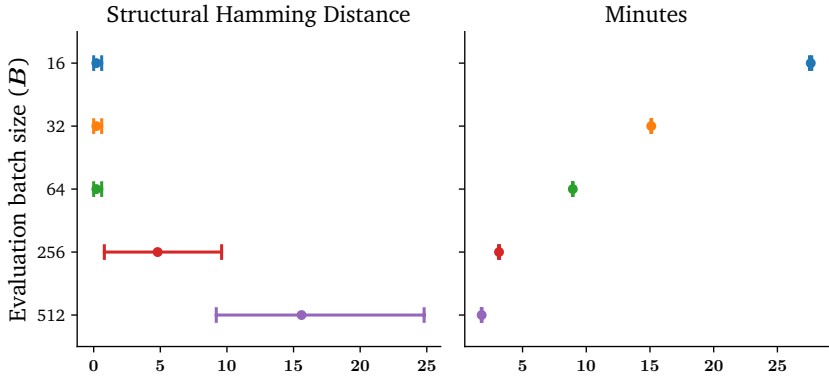

Figure 9: **Effect of Evaluation Batch Size** $B$. We evaluate our method on linear-Gaussian data with 20ER4 graphs and 1,000 observations. Error bars indicate 95% confidence intervals over 5 runs. The number of evaluations is limited to 20,000.

### F.2.4  EFFECT OF NUMBER OF PRELIMINARY CANDIDATES ($C$)

We show the variation our **DrBO**'s performance w.r.t. different numbers of preliminary candidates $C$ in Figure 10, showing that the best performance and runtime can be achieved at 10,000 candidates, and even with 10x more candidates, the runtime of our method increases only slightly.

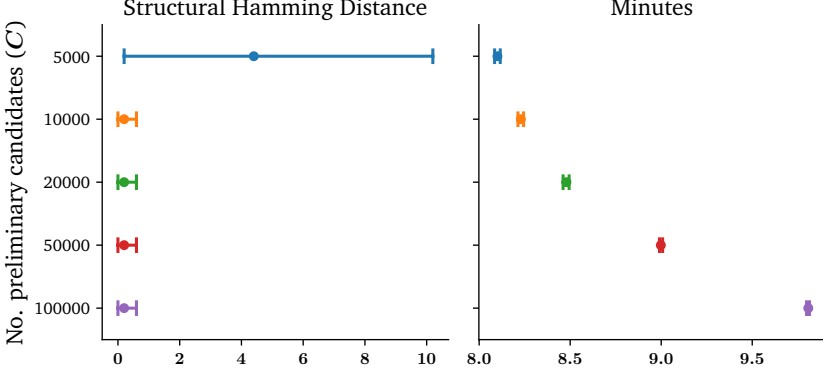

Figure 10: **Effect of Number of Preliminary Candidates** $C$. We evaluate our method on linear-Gaussian data with 20ER4 graphs and 1,000 observations. Error bars indicate 95% confidence intervals over 5 runs. The number of evaluations is limited to 20,000.

### F.2.5 EFFECT OF NUMBER OF TRAINING STEPS PER BO ITERATION ($n_{\text{GRADS}}$)

We study the effect of the number of gradient steps in each BO iteration ($n_{\text{grads}}$) in Figure 11. In general, small values may lead to underfitting and large values may be prone to overfitting, so values in the middle are better for this hyperparameter.

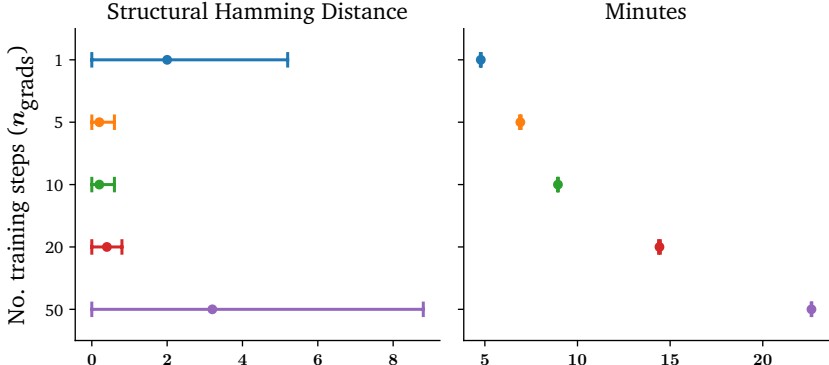

Figure 11: **Effect of Number of Training Steps per BO Iteration** $n_{\textbf{grads}}$. We evaluate our method on linear-Gaussian data with 20ER4 graphs and $1,000$ observations. Error bars indicate $95\%$ confidence intervals over 5 runs. The number of evaluations is limited to $20,000$.

### F.2.6 EFFECT OF REPLAY BUFFER SIZE ($n_{\text{REPLAY}}$)

In Figure 12, we show that higher values for the replay buffer size significantly reduces SHD but does not considerably influence the runtime.

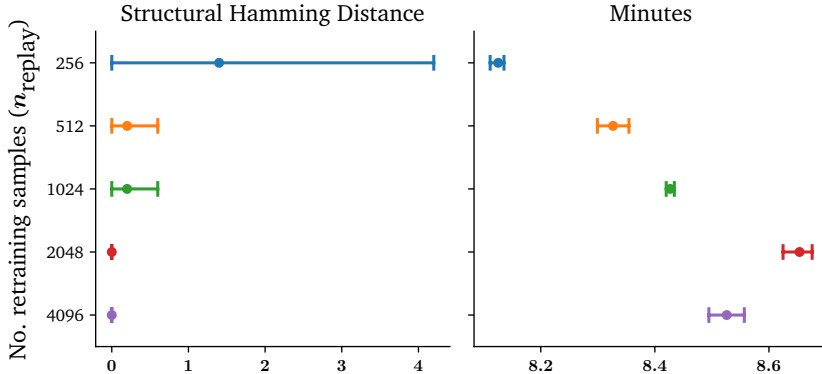

Figure 12: **Effect of Replay Buffer Size** $n_{\textbf{replay}}$. We evaluate our method on linear-Gaussian data with 20ER4 graphs and $1,000$ observations. Error bars indicate $95\%$ confidence intervals over 5 runs. The number of evaluations is limited to $20,000$.

### F.2.7 EFFECT OF LEARNING RATE

Figure 13 depicts that the learning rate has a weak effect on the performance and scalability of our method, where any value below 1 can achieve the same level of SHD and runtime. The SHD only becomes large for a high learning rate of 1.

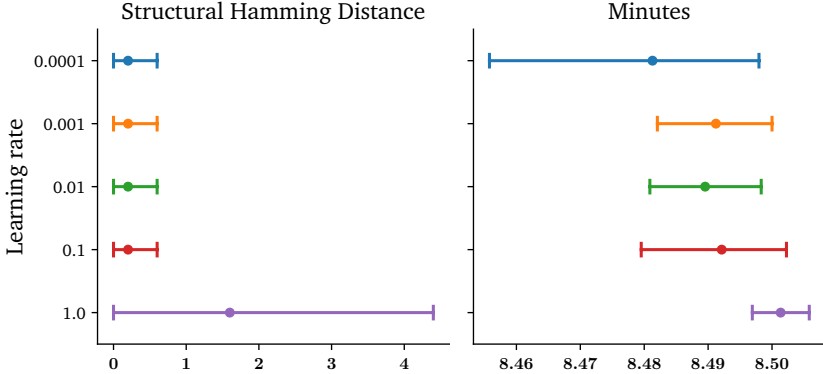

Figure 13: **Effect of Learning Rate.** We evaluate our method on linear-Gaussian data with 20ER4 graphs and 1,000 observations. Error bars indicate 95% confidence intervals over 5 runs. The number of evaluations is limited to 20,000.

### F.2.8    EFFECT OF NUMBER OF HIDDEN UNITS

We present in Figure 14 that the number of hidden units in our dropout networks also has a visible effect on our method's performance, but not much on the runtime. Specifically, a moderate value of 32 achieves a vanishing SHD with the equivalent runtime as others. Meanwhile, to many hidden units may challenge the training process so performance may drop.

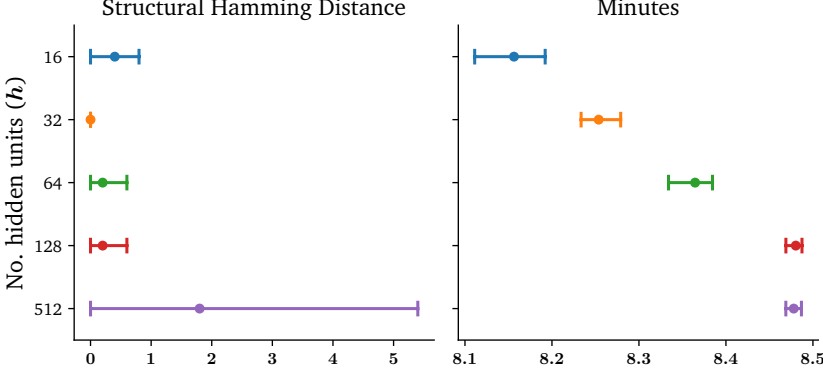

Figure 14: **Effect of Number of Hidden Units** $h$**.** We evaluate our method on linear-Gaussian data with 20ER4 graphs and 1,000 observations. Error bars indicate 95% confidence intervals over 5 runs. The number of evaluations is limited to 20,000.

### F.2.9    EFFECT OF NUMBER OF DROPOUT RATE

Figure 15 suggests that the performance of our method improves with higher dropout rates, while the runtime does not vary significantly.

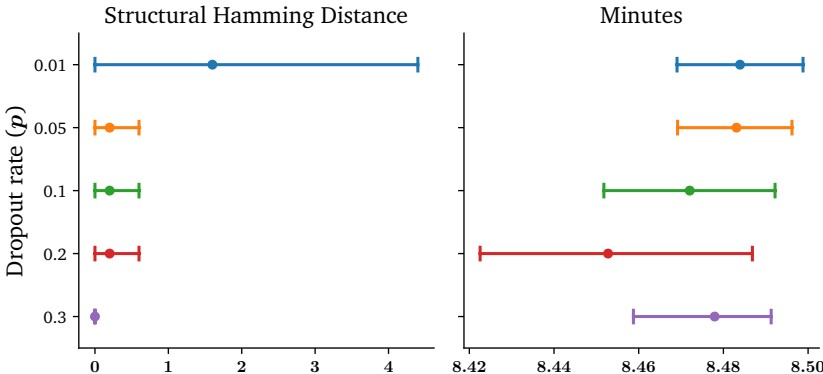

Figure 15: **Effect of Dropout Rate** $p$. We evaluate our method on linear-Gaussian data with 20ER4 graphs and 1,000 observations. Error bars indicate 95% confidence intervals over 5 runs. The number of evaluations is limited to 20,000.

## G    ADDITIONAL BASELINES

Apart from the score-based competitors compared so far, here we also consider additional baselines for comparison, such as conventional methods PC (Spirtes et al., 2000) and GES (Chickering, 2002). In addition, ordering-based methods have also been gaining popularity (Reisach et al., 2021; Rolland et al., 2022; Sanchez et al., 2023; Montagna et al., 2023). These methods sidestep the acyclicity issue by first learning a topological ordering of the causal DAG, then pruning the fully-connected DAG induced by the ordering to obtain the causal structure.

In Table 6, we provide a comprehensive comparison between our method and conventional methods PC and GES, as well as popular ordering-based method, including sortnregress (Reisach et al., 2021), SCORE (Rolland et al., 2022), and DiffAN (Sanchez et al., 2023), on linear, nonlinear, as well as real data. The empirical evaluations indicate that our method is also able to surpass these methods in all metrics and settings.

Table 6: **Comparison with Ordering-based Methods.** We compare our method with conventional methods PC (Spirtes et al., 2000) and GES (Chickering, 2002), as well as popular ordering-based algorithms sortnregress (Reisach et al., 2021), SCORE (Rolland et al., 2022), and DiffAN (Sanchez et al., 2023). The orderings produced by those methods are first transformed into respective fully connected DAGs. Then, for fairness, all DAGs, including ours, are pruned using the weight matrix thresholded at 0.3 for linear data, CAM pruning (Bühlmann et al., 2014) for nonlinear data, and KCIT (Zhang et al., 2011) for the Sachs dataset. The figures are mean ± std over 5 datasets, except for the Sachs dataset. Note that **DrBO**'s result in Figure 1(c) with SHD ≈ 0.4 is obtained without pruning, while using CAM pruning leads to a few missing edges as shown below.

| Dataset | Method | SHD ↓ | FDR ↓ | TPR ↑ | $F_1$ ↑ |
|---|---|---|---|---|---|
| Linear Data (10ER2 graphs, 1,000 samples) | PC | $10.8 \pm 5.1$ | $0.29 \pm 0.1$ | $0.49 \pm 0.2$ | $0.56 \pm 0.2$ |
| | GES | $11.6 \pm 6.4$ | $0.40 \pm 0.2$ | $0.61 \pm 0.2$ | $0.58 \pm 0.2$ |
| | sortnregress | $2.0 \pm 2.6$ | $0.10 \pm 0.1$ | $0.94 \pm 0.1$ | $0.92 \pm 0.1$ |
| | SCORE | $\mathbf{0.0 \pm 0.0}$ | $\mathbf{0.0 \pm 0.0}$ | $\mathbf{1.00 \pm 0.0}$ | $\mathbf{1.00 \pm 0.0}$ |
| | DiffAN | $16.6 \pm 4.9$ | $0.54 \pm 0.1$ | $0.49 \pm 0.1$ | $0.48 \pm 0.1$ |
| | **DrBO** (ours) | $\mathbf{0.0 \pm 0.0}$ | $\mathbf{0.0 \pm 0.0}$ | $\mathbf{1.00 \pm 0.0}$ | $\mathbf{1.00 \pm 0.0}$ |
| Linear Data (30ER8 graphs, 1,000 samples) | PC | $227.0 \pm 8.6$ | $0.44 \pm 0.0$ | $0.10 \pm 0.0$ | $0.17 \pm 0.0$ |
| | GES | | Failed to halt | | |
| | sortnregress | $101.4 \pm 21.8$ | $0.24 \pm 0.0$ | $0.80 \pm 0.1$ | $0.78 \pm 0.0$ |
| | SCORE | $247.0 \pm 11.8$ | $0.76 \pm 0.1$ | $0.05 \pm 0.0$ | $0.08 \pm 0.1$ |
| | DiffAN | $236.0 \pm 3.7$ | $0.58 \pm 0.1$ | $0.16 \pm 0.1$ | $0.23 \pm 0.1$ |
| | **DrBO** (ours) | $\mathbf{1.6 \pm 1.5}$ | $\mathbf{0.00 \pm 0.0}$ | $\mathbf{0.99 \pm 0.0}$ | $\mathbf{1.0 \pm 0.0}$ |
| Nonlinear Data (10ER4 graphs, 1,000 samples) | PC | $32.2 \pm 2.2$ | $0.50 \pm 0.2$ | $0.21 \pm 0.1$ | $0.29 \pm 0.1$ |
| | GES | $29.6 \pm 5.6$ | $0.43 \pm 0.2$ | $0.30 \pm 0.1$ | $0.38 \pm 0.1$ |
| | sortnregress | $27.4 \pm 3.3$ | $0.42 \pm 0.1$ | $0.37 \pm 0.1$ | $0.45 \pm 0.1$ |
| | SCORE | $9.4 \pm 4.6$ | $0.11 \pm 0.1$ | $0.83 \pm 0.1$ | $0.86 \pm 0.1$ |
| | DiffAN | $18.6 \pm 3.8$ | $0.34 \pm 0.1$ | $0.61 \pm 0.1$ | $0.63 \pm 0.1$ |
| | **DrBO** (ours) | $\mathbf{4.2 \pm 1.3}$ | $\mathbf{0.00 \pm 0.0}$ | $\mathbf{0.90 \pm 0.0}$ | $\mathbf{0.95 \pm 0.0}$ |
| Sachs et al. (2005), (11 nodes, 17 edges, 853 samples) | PC | $11$ | $0.25$ | $0.35$ | $0.39$ |
| | GES | $11$ | $0.25$ | $0.35$ | $0.39$ |
| | sortnregress | $13$ | $0.44$ | $0.29$ | $0.38$ |
| | SCORE | $12$ | $0.33$ | $0.35$ | $0.46$ |
| | DiffAN | $16$ | $0.75$ | $0.12$ | $0.16$ |
| | **DrBO** (ours) | $\mathbf{9}$ | $\mathbf{0.11}$ | $\mathbf{0.47}$ | $\mathbf{0.62}$ |

## H   TIME COMPARISONS

Table 7: Runtime comparison on 30ER8 linear-Gaussian data (experiment in Figure 1(a)).

| Method | Max number of evaluations | | | | | |
|---|---|---|---|---|---|---|
| | 10,000 | | 20,000 | | 50,000 | |
| | SHD | Runtime (mins) | SHD | Runtime (mins) | SHD | Runtime (mins) |
| ALIAS | $208.8 \pm 13.1$ | $0.5 \pm 0.0$ | $165.0 \pm 18.0$ | $1.1 \pm 0.0$ | $105.2 \pm 2.9$ | $2.4 \pm 0.0$ |
| CORL | $169.0 \pm 17.1$ | $1.1 \pm 0.0$ | $162.0 \pm 13.4$ | $2.1 \pm 0.0$ | $145.4 \pm 29.3$ | $5.2 \pm 0.1$ |
| COSMO | $213.6 \pm 17.4$ | $0.6 \pm 0.0$ | $203.4 \pm 21.0$ | $1.1 \pm 0.0$ | $195.8 \pm 10.8$ | $2.8 \pm 0.0$ |
| DAGMA | $222.2 \pm 11.2$ | $\mathbf{0.0 \pm 0.0}$ | $218.8 \pm 10.6$ | $\mathbf{0.0 \pm 0.0}$ | $181.6 \pm 22.4$ | $\mathbf{0.0 \pm 0.0}$ |
| **DrBO** (ours) | $\mathbf{2.0 \pm 1.2}$ | $4.6 \pm 0.0$ | $\mathbf{2.0 \pm 1.4}$ | $9.2 \pm 0.0$ | $\mathbf{1.6 \pm 1.5}$ | $22.9 \pm 0.1$ |

Table 8: Runtime comparison on 100ER2 linear-Gaussian data (experiment in Figure 1(b)).

| Method | Max number of evaluations | | | | | |
| --- | --- | --- | --- | --- | --- | --- |
| | 10,000 | | 20,000 | | 50,000 | |
| | SHD | Runtime (mins) | SHD | Runtime (mins) | SHD | Runtime (mins) |
| ALIAS | $230.4 \pm 32.7$ | $2.7 \pm 0.3$ | $136.8 \pm 30.2$ | $5.5 \pm 0.3$ | $32.0 \pm 16.7$ | $13.5 \pm 0.6$ |
| CORL | $148.0 \pm 42.9$ | $11.1 \pm 0.1$ | $120.2 \pm 18.1$ | $22.0 \pm 0.2$ | $81.0 \pm 19.9$ | $54.2 \pm 0.5$ |
| COSMO | $111.0 \pm 10.9$ | $0.8 \pm 0.0$ | $111.8 \pm 12.2$ | $1.5 \pm 0.0$ | $112.6 \pm 13.4$ | $3.7 \pm 0.0$ |
| DAGMA | $124.8 \pm 17.4$ | $0.0 \pm 0.0$ | $93.6 \pm 16.1$ | $0.1 \pm 0.0$ | $6.6 \pm 4.0$ | $0.3 \pm 0.0$ |
| **DrBO** (ours) | $\mathbf{29.2 \pm 16.7}$ | $12.7 \pm 0.1$ | $\mathbf{3.4 \pm 4.3}$ | $25.4 \pm 0.3$ | $\mathbf{1.4 \pm 1.1}$ | $62.7 \pm 0.8$ |

Table 9: Runtime comparison on 10ER4 nonlinear data with Gaussian processes (experiment in Figure 1(c)).

| Method | Max number of evaluations | | | | | |
| --- | --- | --- | --- | --- | --- | --- |
| | 1,000 | | 2,000 | | 20,000 | |
| | SHD | Runtime (mins) | SHD | Runtime (mins) | SHD | Runtime (mins) |
| ALIAS | $15.8 \pm 4.4$ | $4.0 \pm 0.2$ | $12.6 \pm 4.2$ | $5.8 \pm 0.2$ | $4.0 \pm 2.3$ | $8.0 \pm 0.3$ |
| CORL | $10.4 \pm 3.4$ | $20.9 \pm 3.0$ | $9.2 \pm 2.3$ | $26.5 \pm 3.1$ | $8.4 \pm 3.0$ | $29.9 \pm 2.4$ |
| COSMO | $34.8 \pm 2.6$ | $0.3 \pm 0.1$ | $33.0 \pm 3.3$ | $0.4 \pm 0.1$ | $25.6 \pm 4.1$ | $1.8 \pm 0.1$ |
| DAGMA | $40.4 \pm 2.3$ | $0.0 \pm 0.0$ | $38.4 \pm 2.7$ | $0.1 \pm 0.0$ | $34.2 \pm 3.5$ | $0.8 \pm 0.1$ |
| **DrBO** (ours) | $\mathbf{2.2 \pm 1.6}$ | $3.7 \pm 0.8$ | $\mathbf{0.4 \pm 0.5}$ | $3.9 \pm 0.9$ | $\mathbf{0.4 \pm 0.5}$ | $6.1 \pm 1.0$ |

