# OpenReview forum: "Causal Discovery via Bayesian Optimization"
_ICLR.cc/2025/Conference — ICLR 2025 Poster_

### Official Review · Reviewer_ZehJ · 2024-11-02

**Soundness:** 2
**Presentation:** 3
**Contribution:** 2
**Rating:** 6
**Confidence:** 5

**Summary:**

This paper introduces DrBO, a Bayesian Optimization-based framework for efficient and accurate DAG learning from observational data. By leveraging dropout neural networks instead of Gaussian Processes, DrBO addresses scalability issues while integrating uncertainty in score estimation. Empirical results demonstrate DrBO's improved efficiency and accuracy over existing methods across synthetic and real datasets.

**Strengths:**

- Using BO in causal discovery is interesting and novel.
- The paper is very well written.

**Weaknesses:**

see questions.

**Questions:**

1. To my knowledge, the CBO series of papers assumes that the DAG structure is known and primarily focuses on optimizing policies with this prior knowledge. Therefore, these papers may not be directly relevant to the active causal discovery literature. Please revise this in the introduction to reflect the distinction.

2. The synthetic dataset, as first used in NOTEARS, is known to be relatively easy to learn, making pursuit of very low SHD scores less meaningful in recent research, especially the very simple linear gaussian case. How does your method perform on this dataset after standardization, as described in the paper *Beware of the Simulated DAG*?

3. The proposed method appears to be limited to ANMs in causal discovery, which restricts the scope of the paper. It may be more accurate to frame the task as DAG structure learning or Bayesian structure learning rather than causal discovery.

4. In Section 4.1, the authors mention that their method incorporates a low-rank adaptation of Vec2DAG. Does this imply an assumption about the data’s structure, as discussed in *On Low Rank Directed Acyclic Graphs and Causal Structure Learning*? Additionally, what would occur if \( k < d \)?

5. Replacing the Gaussian Process in Bayesian Optimization with Dropout is not uncommon, so it may not warrant being highlighted as a novel contribution in this paper.

6. While many prior works employ CAM as a pruning method, I believe this approach may lack justification here. Why would score-based search methods, including this paper, attempt to prune under nonlinear conditions? It's unusual for newly proposed methods to rely on post-processing from an older method.

7. Please compare this baseline method, *Truncated Matrix Power Iteration for Differentiable DAG Learning*, to your approach.

8. In Figure 1, several methods being compared fail to converge. For the tabulated results, have you ensured that all comparison methods have converged? Additionally, consider moving the running time details from the appendix to the main content, as the high time complexity is a notable limitation of the proposed method.

9. Please provide results for nonlinear functions with datasets of 50 and 100 nodes, detailing both performance metrics and running time.

---

> ### Author Response · Authors · 2024-11-15
> **Response to Reviewer ZehJ (1/3)**
>
> We are greatful for Reviewer ZehJ's invaluable comments. We try our best to address your concerns and we look forward to your responses.
>
> **To my knowledge, the CBO series of papers assumes that the DAG structure is known and primarily focuses on optimizing policies with this prior knowledge. Therefore, these papers may not be directly relevant to the active causal discovery literature. Please revise this in the introduction to reflect the distinction.**
>
> Thank you for your kind suggestion. We would like to clarify that we mentioned them to highlight that our application of BO is entirely different from existing causal discovery studies, since it is easily mistaken that our method is another CBO method, and we have revised our manuscript to be more precise.
>
> **The synthetic dataset, as first used in NOTEARS, is known to be relatively easy to learn, making pursuit of very low SHD scores less meaningful in recent research, especially the very simple linear gaussian case.**
>
> This is not neccessarily the case. We would like to clarify that our dataset involves much denser graphs, and achieving a low SHD in this case is very challenging for existing methods. Specifically, for dense graphs 30ER8, most baselines, especially the *sortnregress* approach introduced in *Beware of the Simulated DAG*, obtains a very high SHD of 100+, while our method's SHD is only 1.6±1.5 (see our Figure 1a and Table 5).
>
> Indeed, this can be confirmed by calculating the Varsortability from *Beware of Simulated DAG*. If Varsortability=1 then the causal structure can be recovered simply by sorting the nodes by increasing variances, i.e., using *sortnregress*. Below, we show the Varsortability (mean±std over 100 simulations) for 30-node graphs with varying densities. The results suggest that while *sortnregress* may be useful for very sparse graphs, it becomes invalid very quickly for denser graph.
>
> | Graph | Varsortability |
> | :---: | :------------: |
> | 30ER1 |   0.95±0.04    |
> | 30ER2 |   0.80±0.11    |
> | 30ER4 |   0.21±0.10    |
> | 30ER6 |   0.03±0.02    |
> | 30ER8 |   0.00±0.00    |
>
> In addition, since Varsortability=0 for 30ER8 graphs, it may be tempting to think that sorting the nodes by decreasing variance can help. To test this, we again apply *sortnregress* with both sorting directions in the Table below (mean±std over 100 simulations).
>
> |               Method               |     SHD      |    Extra    |   Missing   |   Reverse    |
> | :--------------------------------: | :----------: | :---------: | :---------: | :----------: |
> | sortnregress (increasing variance) | 107.09±33.36 | 79.62±24.73 | 15.73±6.98  |  11.74±4.07  |
> | sortnregress (decreasing variance) | 333.78±9.26  | 102.39±8.49 | 106.47±8.37 | 124.92±11.31 |
>
> This result indicates that our synthetic data is not easy and our method's ability to reach low SHDs in such cases is significant.
>
> **How does your method perform on this dataset after standardization, as described in the paper Beware of the Simulated DAG?**
>
> Following your suggestion, we are conducting experiments with standardized data and will update you once they are completed.
>
> **The proposed method appears to be limited to ANMs in causal discovery, which restricts the scope of the paper. It may be more accurate to frame the task as DAG structure learning or Bayesian structure learning rather than causal discovery.**
>
> We would like to clarify that our method is not limited to ANMs. We have demonstrated in Appendix F.1.4 that our method can also handle logistic nonlinearity. In general, our method can be applied to other causal models, as long as a suitable scoring function is provided.

---

> > ### Author Response · Authors · 2024-11-18
> > **Performance on Standardized Data**
> >
> > Dear Reviewer ZehJ,
> >
> > As mentioned in the common thread, we have added the analyses on standardized datasets, as you requested. Overall, our method can still perform really well on both linear and nonlinear standardized data and surpass other methods. Our method still achieves a very low SHD compared with other baselines on standardized linear-Gaussian data, which is attributed mainly by misoriented edges, potentially due to the unidentifiability of the standardized model. Meanwhile, our method still obtains an SHD≈0 on standardized nonlinear data.
> >
> > Please refer to Appendix F.1.5 of our updated manuscript for more details.

---

> ### Author Response · Authors · 2024-11-15
> **Response to Reviewer ZehJ (2/3)**
>
> **In Section 4.1, the authors mention that their method incorporates a low-rank adaptation of Vec2DAG. Does this imply an assumption about the data’s structure, as discussed in On Low Rank Directed Acyclic Graphs and Causal Structure Learning?**
>
> Yes, our study also assumes the low-rank structure, but implicitly. Specifically, like the sparsity assumption that is usually implicitly imposed on the causal structure in existing methods, we did not explicitly outlined it as an assumption in our presentation. Moreover, our experiments did not enforce the low-rank assumption on the synthetic data, showing that our method is not restricted to low-rank structures. The high empirical performance of our method on general graphs of different types (ER, SF, dense, large, and real structures) suggests that our low-rank representation is robust to many kinds of graphs.
>
> **What would occur if ( k < d )?**
>
> When $k<d$, the search space is much lower dimensional, allowing us to potentially reach higher-scoring DAGs faster, as shown in Figure 3a. At the same time, the set of DAGs representable by the low-rank representation is reduced from the set of all DAGs, potentially excluding the ground truth DAG. However, our experiments using only rank $k=8$ have shown that our method can still attain a very low SHD even for much larger $d\in\\{30,100\\}$ (see our Figure 1a and 1b). In addition, in Figure 3a, we have empirically analyzed the effect of different ranks and found that even a low rank $k=2$ can still result in a near-zero SHD for complex 30ER8 graphs that have dense connections.
>
> Furthermore, due to the curse of dimensionality, lower ranks, which associate with lower-dim representations, allow for generating more unique candidates compared with higher ranks, so we can more likely meet the higher-scoring DAGs earlier during exploration. We empirically test this by calculating the number of unique DAGs among 1000 random DAGs generated using different ranks in the Table below (the numbers are mean±std over 10 simulations).
>
> |Rank|Number of unique 30-node DAGs over 1000 random DAGs|
> |:-:|:-:|
> |k=2 (90 dims)|926.7±7.0|
> |k=4 (150 dims)|779.2±12.7|
> |k=8 (270 dims)|493.5±12.3|
> |k=12 (390 dims)|332.4±10.8|
> |k=32 (990 dims)|90.7±9.5|
>
> This shows that the lower the rank, the more unique DAGs we can consider for exploration. For $k=2$, almost every DAG among 1000 generated DAGs is unique, whereas the lowest number of unique DAGs of less than 10% of all DAGs is obtained using $k=32\approx d$.
>
> **Replacing the Gaussian Process in Bayesian Optimization with Dropout is not uncommon, so it may not warrant being highlighted as a novel contribution in this paper.**
>
> We would like to clarify that, as outlined in the "Contributions" part of the Introduction, we did not highlight replacing GPs with dropout networks as a novelty in our paper. Rather, we mentioned it as one of the "key design choices", highlighting it as a neccessary leverage to enable BO in causal discovery, which is our novel contribution in the context of causal discovery.
>
> **While many prior works employ CAM as a pruning method, I believe this approach may lack justification here. Why would score-based search methods, including this paper, attempt to prune under nonlinear conditions? It's unusual for newly proposed methods to rely on post-processing from an older method.**
>
> We would like to clarify that we employed only the *pruning step* from CAM, not the whole "CAM as pruning method". In CAM pruning, each variable is regressed against its parents in the raw estimated DAG, which may contain many extra edges due to overfitting, by using generalized additive model regression, then insignificant parents are removed. This is itself a reasonable method for pruning extra edges in additive models because it can approximate the data generation process, and thus is widely employed in not just score-based methods (GraN-DAG, RL-BIC, ALIAS, etc.), but also ordering-based methods (e.g., CORL, SCORE, DiffAN, etc.), where pruning is required to remove extra edges in the fully-connected DAGs induced by the returned causal orderings.
>
> **Please compare this baseline method, Truncated Matrix Power Iteration for Differentiable DAG Learning, to your approach.**
>
> Following your suggestion, we have added this baseline to the main experiments in our revision. We used NOTEARS's implementation combined with TMPI DAG constraint and call the method 'NOTEARS+TMPI' in our experiments. This method's performance is relatively strong to some other baselines, especially in sparse graphs (Figure 1b and Table 1 of the revised manuscript) and nonlinear data (Figure 1c and 2 of the revised manuscript), however, it is still surpassed by our method in all cases.

---

> ### Author Response · Authors · 2024-11-15
> **Response to Reviewer ZehJ (3/3)**
>
> **In Figure 1, several methods being compared fail to converge. For the tabulated results, have you ensured that all comparison methods have converged?**
>
> We would like to clarify that, to demonstrate the sample-efficiency of our approach, it is our deliberate choice to show that the baselines fail to converge before our method. This is because the number of optimization steps, or more generally, the computational budget, is an important hyperparameter in score-based methods that strongly influences the causal discovery performance, but is usually overlooked. Therefore, our empirical study aims to control for this factor, and thus our Figure 1 is intended to show that our method can converge to low SHDs using fewer steps than other baselines, when they have not converged and their SHDs are still high.
>
> To further show that other methods can indeed converge when given more optimization steps, the Table below contains the converged performance of all methods for the case of linear data with 10 nodes (mean±std over 10 simulations), in which 5/6 baselines can converge to a near-zero SHD. COSMO converges to a non-zero SHD, which is in accordance with the reported results in their paper.
>
> | Graph | Method       | SHD     | FDR     | TPR     | F1      |
> | :---- | :----------- | :------ | :------ | :------ | :------ |
> | ER-1  | ALIAS        | 0.0±0.0 | 0.0±0.0 | 1.0±0.0 | 1.0±0.0 |
> | ER-1  | CORL         | 0.0±0.0 | 0.0±0.0 | 1.0±0.0 | 1.0±0.0 |
> | ER-1  | COSMO        | 1.7±1.8 | 0.1±0.1 | 0.9±0.1 | 0.9±0.1 |
> | ER-1  | DAGMA        | 0.5±1.1 | 0.0±0.0 | 0.9±0.3 | 1.0±0.0 |
> | ER-1  | NOTEARS+TMPI | 0.5±1.6 | 0.0±0.1 | 1.0±0.1 | 1.0±0.1 |
> | ER-1  | GOLEM        | 0.0±0.0 | 0.0±0.0 | 1.0±0.0 | 1.0±0.0 |
> | ER-2  | ALIAS        | 0.1±0.3 | 0.0±0.0 | 1.0±0.0 | 1.0±0.0 |
> | ER-2  | CORL         | 0.1±0.3 | 0.0±0.0 | 1.0±0.0 | 1.0±0.0 |
> | ER-2  | COSMO        | 5.0±3.3 | 0.2±0.1 | 0.9±0.1 | 0.8±0.1 |
> | ER-2  | DAGMA        | 0.8±1.5 | 0.0±0.0 | 1.0±0.1 | 1.0±0.0 |
> | ER-2  | NOTEARS+TMPI | 0.6±1.9 | 0.0±0.1 | 1.0±0.0 | 1.0±0.0 |
> | ER-2  | GOLEM        | 0.2±0.6 | 0.0±0.0 | 1.0±0.0 | 1.0±0.0 |
>
> **Additionally, consider moving the running time details from the appendix to the main content, as the high time complexity is a notable limitation of the proposed method.**
>
> Regarding runtime, in the main paper we already have the runtime analysis wrt. performance (last column of Figure 1), showing that our method can arrive at near-zero SHD with less time than other methods, and the runtime details in Appendix H are merely numerical supplementary.
>
> In addition, we believe that when assessing a causal discovery method, running time should be accompanied with a performance metric like SHD, which is the primary measure of causal discovery accuracy, since a method can be very fast and still achieve poor performance. Our results in Figure 1, especially the last two columns, have demonstrated that within a given budget (either number of DAG evaluations or runtime), our method usually achieves better performance than other methods, and given more budget, our method can arrive at very accurate DAGs, while many methods still struggle. For example, in 5 minutes, while our method has achieved a very low SHD=2 for the highly challenging graphs 30ER8, the second-best method is still at SHD=16, and the remaining baselines are still very far behind with SHD>100. This trend can also be observed for nonlinear data, when our method can reach SHD≈0 in 5 minutes, when other methods are still at SHD>6.
>
> **Please provide results for nonlinear functions with datasets of 50 and 100 nodes, detailing both performance metrics and running time.**
>
> We are conducting these experiments and will update once they are completed.
>
> In the meantime, we look forward to further discussions with you to resolve any outstanding issues.

---

> > ### Author Response · Authors · 2024-11-21
> > **Update on Large-scale Nonlinear Experiments**
> >
> > Dear reviewer ZehJ,
> >
> > We sincerely thank you again for your insightful comments. In the latest revision, we have conducted additional analyses on nonlinear datasets with 50 and 100 nodes, addressing both performance and runtime, as per your request. The detailed results can be found in Appendix F.1.7. For your convenience, we provide a summary of the results in the table below:
> >
> > | **Nodes** | **Method**      | **Minutes to reach SHD=20** | **Minutes to reach SHD=10** | **Minutes to reach SHD=5** | **Minutes to reach SHD=2** |
> > |-----------|-----------------|-----------------------------|-----------------------------|----------------------------|----------------------------|
> > | 50        | ALIAS           | 1.8 ± 0.6                   | 5.1 ± 0.9                   | 8.3 ± 2.5                  | 9.3 ± 3.1                  |
> > |           | **DrBO (ours)** | **1.1 ± 0.5**               | **2.1 ± 1.4**               | **4.5 ± 3.2**              | **6.8 ± 4.2**              |
> > | 100       | ALIAS           | 6.5 ± 2.6                   | 16.3 ± 2.5                  | 24.0 ± 4.5                 | 33.7 ± 6.1                 |
> > |           | **DrBO (ours)** | **2.9 ± 0.6**               | **5.0 ± 1.3**               | **8.0 ± 3.3**              | **12.2 ± 3.9**             |
> >
> > As shown in the table, our method achieves low SHDs within minutes even on nonlinear datasets with 50 and 100 nodes. Furthermore, it surpasses the baseline method in both accuracy and runtime, and is particularly faster in attaining lower SHDs, highlighting the efficiency and scalability of our approach.
> >
> > Having addressed your questions and conducted these additional experiments in response to your feedback, we kindly request that you consider our responses and re-evaluate our submission. We greatly appreciate your time and thoughtful evaluation.
> >
> > Thank you again for your valuable input!

---

> > > ### Comment · Reviewer_ZehJ · 2024-11-22
> > > **Thank you**
> > >
> > > Thank you for addressing my concerns. While, actually, I also have the concern that the results seem almost too good to be true. (No worries, I am mot concerning about the correctness of the results). This raises the question of whether it’s time to move beyond using simulation data for testing causal discovery methods. The field may benefit from tackling more challenging, real-world problems. Alternatively, I would encourage the authors to openly discuss this perspective and elaborate on what they see as the core challenges moving forward. Overall, I appreciate the detailed response and have raised my score to 6.

---

> > ### Comment · Reviewer_ZehJ · 2024-11-27
> > **Unfair comparisons**
> >
> > As I previously mentioned, the compared methods, such as DAGMA, do not converge under the given conditions. The authors claim that all methods can be compared using the same "maximum number of evaluations." However, this approach is problematic, as different methods employ different optimization strategies. For example, the evaluation at 50000 steps indicates that DAGMA has not yet converged.
> >
> > Running time could serve as an alternative metric for comparison. However, I find discrepancies in the results presented:
> >
> > In Figure 1, it is stated that DAGMA runs for 5 minutes without converging. This contradicts my own experience, where DAGMA typically converges in about 10 seconds.
> > While DAGMA may require more steps to converge—potentially around 150000 or 200000 steps—it still demonstrates significantly lower runtime compared to the proposed methods. For instance, DAGMA takes approximately 1 minute to converge, whereas the proposed methods require about 60 minutes for datasets of similar scale.
> >
> > These comparisons appear fundamentally unfair. I suspect the authors are aware of these issues but have chosen not to disclose this information transparently.

---

> ### Author Response · Authors · 2024-11-27
> **Rest assured that our comparison is fair**
>
> Dear Reviewer ZehJ,
>
> Thank you for bringing this to our attention!
>
> We assure you that our comparison remains fair.
>
> We would like to clarify that, we did not limit DAGMA to just 50k steps in the last column of Figure 1, but in fact, we limited it to millions of evaluations, and we removed its early stopping condition to allow it to run for an arbitrary amount of time, so its cut-off runtime in the last column in Figure 1 can be equal to all other methods.  Without this, we cannot answer the question "what is the performance of each method if it is run for x steps / y minutes?". We have also mentioned this in Appendix E: "Regarding the number of evaluations, for all methods, we run more than needed then cut off at common thresholds."
>
> Specifically, for DAGMA, we run it for around 24 million steps by setting T = 800 (the default is T = 5), so that we can either use a threshold at 50k evaluations or at 5 mins with the full learning progression of the method. In deed, DAGMA can reach 50k steps in just a few seconds as usual, but costs around 5 million steps at 5 mins for 30ER8 data.
>
> We strongly believe that our comparison is reasonable for controling for both the number of evaluations and runtime, and we will most certainly clarify this better in the revision.
>
> In light of this clarification, we hope you can reconsider your assessment, and we are open to further discussions regarding any outstanding concern. Thank you again for your time!

---

> ### Author Response · Authors · 2024-11-28
> **Updated revision**
>
> Dear Reviewer ZehJ,
>
> In our last response, we have clarified that DAGMA runs for minutes in our study instead of just a few seconds is simply because we allowed it to, so that we can answer the question of *How good a method can become within a given runtime?*.
>
> In addition to that, we have uploaded a revision clarifying your concerns.
>
> Particularly, in Appendix D.4, we have included:
>
> > Additionally, in this study, we evaluate the performance of various methods with respect to both the number of steps and runtime, addressing two independent questions: “How accurate can a method become given a fixed number of steps?” and “How accurate can it be within a given runtime?”. To ensure fair comparisons, our second question accounts for potential biases in measuring performance solely by the number of DAG evaluations. This is particularly important for methods like gradient-based approaches (e.g., DAGMA), which may require many steps but still exhibit low overall runtime.
>
> > To address this, we use runtime as a more equitable efficiency metric. Specifically, we set a high number of steps for all methods (e.g., we use T=800 iterations instead the default of only T=5 for DAGMA on linear data) and disable early stopping if applicable, to capture their progression over an extended period of time. We then truncate the tracking data, which contains performance metrics and timestamp at every step, either at a fixed number of steps or a specified runtime, as illustrated in Figure 1. This ensures that the results in the last column of Figure 1 are not constrained by the number of steps. For instance, at the 5-minute mark in the last column of Figure 1a, DAGMA completes approximately 5 million steps compared to only 50,000 steps in the third column.
>
> We have taken every possible measure to ensure fairness in our evaluations and strongly discourage any form of misconduct. In light of our clarification, we would greatly appreciate it if you could reconsider your evaluation.
>
> Thank you sincerely for your time and effort in assessing our paper!

---

> > ### Comment · Reviewer_ZehJ · 2024-11-28
> > **Follow Up**
> >
> > Thank you for the further clarification! I apologize for my previous comment regarding the unfair comparison, which, as the authors have explained, is indeed fair within the current setup.
> >
> > I ran DAGMA myself last night and confirmed that the reported results are correct. I apologize again for questioning the impressive performance comparison, where DrBO achieves an SHD of 1.4, compared to other methods that typically result in SHD values greater than 100. I now realize that the weaker performance I initially observed is due to the limitations of continuous optimization methods, such as NOTEARS, DAGMA, and GOLEM, when handling very dense graphs, like ER8. It's reassuring to see that DrBO can effectively handle such cases.
> >
> > For sparser graphs, like ER1 and ER2, continuous optimization methods tend to converge quickly and yield reasonable results. However, DrBO typically requires much more time to achieve similar results, as shown in Figure 1b and Tables 8 and 9.
> >
> > After reviewing the results again, I acknowledge the advantages of the proposed method and have increased my score to 6.

---

> > > ### Author Response · Authors · 2024-11-28
> > > **Thank you!**
> > >
> > > Thank you, Reviewer ZehJ, for your encouraging feedback!
> > >
> > > We agree that gradient-based methods are very effective for sparse graphs. However, they can struggle to improve performance on more complex structures. Our approach prioritizes accuracy, aiming to tackle arbitrarily complex scenarios, albeit with some trade-off in computational overhead, of course. We hope our work serves as a foundation for future methods that can integrate and balance these objectives, combining their strengths to achieve even better and faster results.
> > >
> > > Once again, thank you for taking the time to carefully review and verify our findings!

---

### Official Review · Reviewer_VYj5 · 2024-11-04

**Soundness:** 2
**Presentation:** 3
**Contribution:** 3
**Rating:** 6
**Confidence:** 3

**Summary:**

This paper develops a Bayesian optimization method for score-based causal discovery. Several design choices are adopted, which include (1) developing a low-rank DAG representation, (2) replacing Gaussian process in conventional Bayesian optimization with dropout neural networks, (3) learning the DAG score indirectly via node-wise local scores, and (4) training in a continual way. Empirical studies are provided.

**Strengths:**

- The paper is well written and easy to follow.
- Developing effective search procedure for score-based causal discovery is an interesting and important topic. The proposed method adopts various design choices and is practical.
- The search method is reasonable.
- The empirical studies demonstrate that the proposed method considerably outperforms existing methods.

**Weaknesses:**

- Some of the baselines considered are not adequate.
- Some of the results may seem too good to be true. For example, achieving a SHD of 1.6 with only 1000 samples across 30 nodes and 240 edges seems highly challenging due to finite sample error. This concern is especially relevant when dealing with nonlinear data. (I look forward to the authors' clarification/explanation on this, and please correct me if I misunderstood anything.)

**Questions:**

- Is there a reason why the paper considers only identifiable models? That is, why general linear Gaussian model cannot be learned by the BIC-NV score?
- Although BIC-NV is given, it seems that the experiments focus on equal variances. I would suggest adding experiments for different variances as well.
- Baselines: For linear case, GOLEM may also be included. Also, adding the results for more conventional search methods, such as GES/FGES, may also be helpful.
- Why does DAGMA-MLP performs so poorly for nonlinear data? The TPR is close to 0. If the reason is due to instability in optimization, the paper may consider adding NOTEARS-MLP that may be more stable.
- For Section 5.2, specifically Sachs data, did the paper use linear or nonlinear version of the method?

---

> ### Author Response · Authors · 2024-11-15
> **Response to Reviewer VYj5**
>
> We thank Reviewer VYj5 for the valuable insights. We address your concerns in the rebuttal below.
>
> **Some of the baselines considered are not adequate**
>
> Following your suggestion, we have enhanced the baselines considerably. Our revised manuscript has incorporated several additional baselines, including GOLEM and NOTEARS with TMPI constraint (see Figure 1, 2, and Table 1). These gain better performance than other baselines in several cases, e.g., they both achieve slightly better performance than DAGMA for large graphs (Figure 1b), and much lower SHDs compared with other baselines for real-world structures (Table 1). In addition, NOTEARS+TMPI also obtains the third-best performance with SHD≈7.2 for nonlinear data with GPs (Figure 1c), and we have also improved DAGMA's performance significantly in this setting, from SHD≈28 to SHD≈17 (Figure 1c).
>
> **Some of the results may seem too good to be true. For example, achieving a SHD of 1.6 with only 1000 samples across 30 nodes and 240 edges seems highly challenging due to finite sample error. This concern is especially relevant when dealing with nonlinear data. (I look forward to the authors' clarification/explanation on this, and please correct me if I misunderstood anything.)**
>
> Thank you for recognizing the hardness of our data setting. We firmly assure you that this accuracy is possible with our method, thanks to BO's ability to effectively optimize the DAG score. Additionally, this level of accuracy has also been achieved in ALIAS, even though with far many more DAG evaluations than ours. To further prove this, we have attached our source code in the Supplementary, so that you can reproduce our results.
>
> **Is there a reason why the paper considers only identifiable models? That is, why general linear Gaussian model cannot be learned by the BIC-NV score?**
>
> We would like to clarify that we considered both identifiable and unidentifiable models in our experiments. While we examined identifiable models in the main paper, we have also studied two unidentifiable causal models in the Appendix, namely the general linear models (without equal noise variance) in Appendix F.1.3 and logistic models in Appendix F.1.4, where our method still works well and can find the highest scores with lowest structural errors most of the time.
>
> **Although BIC-NV is given, it seems that the experiments focus on equal variances. I would suggest adding experiments for different variances as well.**
>
> We would like to clarify that our experiments examined both equal and non-equal variances settings. Specifically, our non-linear experiments (Figure 1c and 2) are for non-equal variances, in which BIC-NV is used for scoring and optimization. We have mentioned in Section 5.1.2 that the noise variances for GP data are sampled uniformly in $[0.4, 0.8]$, and for the real Sachs dataset, the noises are unlikely to have equal variances. Our revised Section 5.1.2 has further clarified this.
>
> **Baselines: For linear case, GOLEM may also be included. Also, adding the results for more conventional search methods, such as GES/FGES, may also be helpful.**
>
> Thank you for your kind recommendation. We have incorporated GOLEM results for linear data in Figure 1 and Table 1 of the revised manuscript. Regarding conventional search methods like GES/FGES, since they are usually shown to perform poorly in many recent studies, and to avoid cluttering the presentation with too many baselines, we did not include them. However, following your suggestion, we will add them in the next revision.
>
> **Why does DAGMA-MLP performs so poorly for nonlinear data? The TPR is close to 0. If the reason is due to instability in optimization, the paper may consider adding NOTEARS-MLP that may be more stable.**
>
> Thank you for your kind suggestion. We have run NOTEARS-MLP for nonlinear data and obtained an SHD of 11.8±2.59. We found that NOTEARS-MLP set the $\ell_1$ and $\ell_2$ weights to zero, while DAGMA-MLP used $\ell_1=0.02$ and $\ell_2=0.005$, leading it to predict very sparse graphs and thus resulting in low TPR. Therefore, we have updated our Figure 1c with $\ell_1=\ell_2=0$ for DAGMA, in which it achieves a significantly higher performance with SHD≈17. Nevertheless, we have also included NOTEARS-MLP in conjunction with TMPI constraint to achieve an SHD≈7 (see our Figure 1c of the revised manuscript).
>
> **For Section 5.2, specifically Sachs data, did the paper use linear or nonlinear version of the method?**
>
> As mentioned in our Appendix E, we used the nonlinear version of our method with GP regression and BIC-NV scoring for the Sachs data, as with all considered baselines.
>
> We hope this rebuttal sufficiently addresses your concerns, and we look forward to further discussions to resolve any outstanding issues.

---

> > ### Comment · Reviewer_VYj5 · 2024-11-25
> >
> > Thanks for the detailed responses and additional experiments. Most of my concerns have been addressed. I have updated my rating from 5 to 6.

---

### Official Review · Reviewer_UyS6 · 2024-11-04

**Soundness:** 3
**Presentation:** 3
**Contribution:** 2
**Rating:** 6
**Confidence:** 3

**Summary:**

This paper proposes an efficient causal discovery algorithm with Bayesian optimization. In particular, the authors consider a variant of Vec2DAG (Duong et al., 2024) for the DAG constraint, and use dropout neural networks and continual training scheme to optimize the adjacency matrix. Experiments show the efficiency and effectiveness of their method.

**Strengths:**

- This paper is written clearly, with clear and detailed descriptions of their method and experiments.

- They performed extensive experiments for validation.

**Weaknesses:**

- While there exist some causal discovery algorithms with Bayesian optimization, it seems not proper to state “To our knowledge, this is the first score-based causal discovery method based on BO ”. I think it should be corrected.

- Throughout the paper, from the experiments, it is demonstrated that the proposed method can give better performances in both accuracy, sample-efficiency, and scalability, compared with other SOTA baselines. Generally, such a great method needs more assumptions or conditions to be satisfied. But intuitively, I cannot find these assumptions or conditions. Did this method have some other implied assumptions or conditions (like more hyperparameters)?

- In experiments, it would be good to compare some Bayesian causal discovery methods (Deleu et al., 2022: Tranet al., 2023; Annadani et al., 2023), since they are all causal discovery methods. Or explain the reasons why not comparing with them.

- The code is not available for reproduction.

**Questions:**

- In Eq.(4), is the matrix $R$ strictly upper-triangular?
- In Figure 1(b), did the authors still use 1000 samples for the large-graph experiments? $n=1000$ for the graph with 100 nodes?
- In Figure 3(a), why in general smaller $k$ could obtain higher performances, compared with the full-rank cases?

---

> ### Author Response · Authors · 2024-11-15
> **Response to Reviewer UyS6**
>
> We thank Reviewer UyS6 for your positive evaluation. Your concerns are addressed in the rebuttal below.
>
> **While there exist some causal discovery algorithms with Bayesian optimization, it seems not proper to state “To our knowledge, this is the first score-based causal discovery method based on BO ”. I think it should be corrected.**
>
> Thank you for your kind comment. We have revised the manuscript to highlight that our study is the first score-based causal discovery method based on BO *for purely observational data*.
>
> **Throughout the paper, from the experiments, it is demonstrated that the proposed method can give better performances in both accuracy, sample-efficiency, and scalability, compared with other SOTA baselines. Generally, such a great method needs more assumptions or conditions to be satisfied. But intuitively, I cannot find these assumptions or conditions. Did this method have some other implied assumptions or conditions (like more hyperparameters)?**
>
> We would like to clarify that our method does not require any additional assumptions compared with existing ones. Specifically, our assumptions outlined in Section 3.2 include causal sufficiency, causal minimality, and identifiable models. These assumptions are standard and are similar to many studies, e.g., NOTEARS, DAGMA, RL-BIC, CORL, etc. Our scoring functions and evaluation data are also the same as RL-BIC, CORL, ALIAS, etc.
>
> Our method is better than the baselines thanks to the effectiveness of BO, in which we predict the scores of DAG candidates to prioritize the DAGs that are most likely to have higher scores, before actually exploring/evaluating them. This helps us avoid examining non-informative DAGs, e.g., DAGs that we are certain to have low scores, and at the same time reveals higher-scoring DAGs earlier. Meanwhile, most existing methods do not effectively take into account past exploration data to make an informed candidate selection, thus resulting in more unneccessary trials than our method.
>
> In addition, our code is now available in the Supplementary material, which can be used to confirm our strong performance.
>
> **In experiments, it would be good to compare some Bayesian causal discovery methods (Deleu et al., 2022: Tranet al., 2023; Annadani et al., 2023), since they are all causal discovery methods. Or explain the reasons why not comparing with them.**
>
> We would like to explain that we did not compare with them for several reasons. Firstly, our study focuses on score-based causal discovery with an emphasis on sequential optimization, which revolves around solving the optimization problem in Eq. (1), so Bayesian causal discovery methods do not directly fall into the same setting as ours. Secondly, it is not very common in the literature for a point-estimate causal discovery study to compare with Bayesian causal discovery studies, so we simply followed common practice. However, following your suggestion, we will certainly add them in the next revision.
>
> **The code is not available for reproduction.**
>
> We have made our code available for reproduction in the Supplementary material.
>
> **In Eq.(4), is the $R$ matrix strictly upper-triangular?**
>
> No, $R$ does not need to be strictly upper-triangular.
>
> **In Figure 1(b), did the authors still use 1000 samples for the large-graph experiments? $n=1000$ for the graph with 100 nodes?**
>
> Yes, we used 1000 samples for all main experiments. We hope this clarification helps you find the performance of our method significant. You can confirm our results with the code provided.
>
> **In Figure 3(a), why in general smaller $k$ could obtain higher performances, compared with the full-rank cases?**
>
> In short, this is because it is much more challenging to search in a very high-dimensional space compared to a lower-dim one. Specifically, for full-rank cases, the search space is much larger and sparser than the low-rank ones. Due to the curse of dimensionality, sampling the same number of random DAG candidates in the full-rank search space tends to lead to fewer unique candidates compared with a low-dim one, reducing the chance to meet the optimal solution earlier.
>
> To empirically verify this, we calculate the number of unique DAGs among 1000 random 30-node DAGs generated with different ranks in the Table below (the numbers are mean±std over 10 simulations).
>
> |Rank|Number of unique DAGs over 1000 random DAGs|
> |:-:|:-:|
> |k=2 (90 dims)|926.7±7.0|
> |k=4 (150 dims)|779.2±12.7|
> |k=8 (270 dims)|493.5±12.3|
> |k=12 (390 dims)|332.4±10.8|
> |Full-rank (465 dims)|421.9±13.8|
> |k=32 (990 dims)|90.7±9.5|
>
> It can be seen that, typically, the lower the rank, the more unique DAGs we can pre-examine for exploration. For k=2, almost every DAG among 1000 generated DAGs is unique, whereas the full-rank representation is higher-dim and can only generate fewer than half the unique DAGs.
>
> We hope this rebuttal addresses your concerns, and are open for further discussions to resolve your remaining concerns.

---

### Official Review · Reviewer_gMGJ · 2024-11-07

**Soundness:** 3
**Presentation:** 3
**Contribution:** 3
**Rating:** 8
**Confidence:** 3

**Summary:**

The authors propose DrBO (DAG recovery via Bayesian Optimization)—a novel DAG learning framework leveraging Bayesian optimization (BO) to find high-scoring DAGs.  To address the scalability issues of conventional BO in DAG learning,  the authors replace Gaussian Processes commonly employed in BO with dropout neural networks, trained in a continual manner. DrBO is computationally efficient and can find the accurate DAG in fewer trials and less time than existing state-of-the-art methods. This is demonstrated through an extensive set of empirical evaluations on many challenging settings with both synthetic and real data.

**Strengths:**

Learning DAG from data using BO is novel and interesting.  The authors overcome the scalability issue of conventional BO by leveraging dropout in neural networks. Experimental results show that the proposed method is effective and can achieve improved results. The paper was written with technical details.

**Weaknesses:**

N/A

**Questions:**

Could the authors give more details on how to ensure the binary adjacent matrix is a DAG in the optimization steps?

---

> ### Author Response · Authors · 2024-11-15
> **Response to Reviewer gMGJ by Authors**
>
> We wholeheartedly thank Reviewer gMGJ for the positive assessment. Below we try our best to address your question.
>
> **Could the authors give more details on how to ensure the binary adjacent matrix is a DAG in the optimization steps?**
>
> In short, the binary adjacency matrix is always ensured to be a DAG thanks to our DAG representation explained in Section 4.1.
>
> In more details, let us first recall our DAG representation:
>
> $$\tau(\bf{p},\bf{R})=H(\mathrm{grad}(\bf{p}))\odot H(\bf{R}\cdot \bf{R}^\top)$$
>
> The acyclicity of this binary matrix is enforced through the first term $H(\mathrm{grad}(\bf{p}))$. This term is an adjacency matrix of a directed graph where there is an edge $i\rightarrow j$ if and only if $p_i < p_j$. By contradiction, assuming there is a directed cycle $i_1\rightarrow i_2\rightarrow\ldots\rightarrow i_1$ in this graph, then it must be that $p_1 < p_1$, which is always impossible. Therefore, there cannot be any directed cycle in this graph, rendering it a DAG. The second term in the equation above, $H(\bf{R}\cdot \bf{R}^\top)$, plays the role of a selection matrix that chooses some edges in the DAG induced by the first term to include in the final result, so there is no new directed cycle and thus the obtained binary matrix is a DAG.

---

> > ### Comment · Reviewer_gMGJ · 2024-11-25
> >
> > Thank you for the clarification. I will keep the score unchanged.

---

### Author Response · Authors · 2024-11-15
**Response to all Reviewers**

We sincerely thank all the reviewers for their time and effort in thoroughly reviewing our manuscript, as well as for providing valuable feedback and comments. We have worked diligently to address the raised issues and look forward to engaging in constructive and meaningful discussions.

We also appreciate the reviewers for recognizing the positive aspects of our submission and acknowledging the novelty and efficiency of our Bayesian Optimization approach to score-based causal discovery. Additionally, we are grateful for the recognition of our effective presentation and comprehensive experimental analysis.

We have addressed each review in the respective threads, and in response to your kind requests, we have made several revisions and additions to our manuscript. The major changes of our manuscript are the inclusion of additional strong and relevant baselines, including:
- GOLEM for linear data (Figure 1a, 1b, and Table 1).
- NOTEARS with TMPI constraint (Figure 1, 2, and Table 1).

These new baselines improve upon the previous ones in several cases, however, they are still visibly outperformed by our method, thus further strengthening the significance of our empirical evaluations. We kindly request the reviewers to refer to the updated version of the manuscript for further discussions.

In addition, we have also attached our source code for reproduction in the Supplementary material, so that you can confirm the strong performance of our method.

---

> ### Author Response · Authors · 2024-11-18
> **Update: Experiments on Standardized Data**
>
> Dear all reviewers,
>
> In our recent revision, we have included a new section (**Appendix F.1.5**) to delve deeper into how our method performs on standardized data, which may makes causal discovery less trivial, as discussed in the study *Beware of the Simulated DAG!*. We have found that **our method continues to perform really well on both linear and nonlinear standardized datasets**, further solidifying its reliability:
> - For linear-Gaussian data, standardization renders the causal model unidentifiable, thus making it impossible to consistently recover the correct causal structure. Yet, the empirical results reveal that our method still significantly outperforms other baselines with very low errors (SHD≈3 for 10ER2 graphs, while the second-best SHD is ≈20). Specifically, our method barely produces any missing or extra edge and only mis-orients a few edges due to the unidentifiability of the causal model.
> - For nonlinear ANM data, since the causal model remains identifiable after standardization, the causal discovery result is similar to our experiment on the same datasets without standardization. More particularly, our method can still accurately recover the causal structures with an SHD≈0.
>
> For the detailed analyses, please refer to Appendix F.1.5 of our latest revision.

---

### Author Response · Authors · 2024-11-21
**Discussion reminder**

Dear all reviewers,

We would like to once again express our deep gratitude for your valuable time and effort in reviewing our manuscript. In response to your insightful comments, we have extensively addressed your concerns and questions and have significantly revised the manuscript to reflect them. **For your convenience, the revisions have been clearly marked in red with margin notes.**

To enhance the clarity of our manuscript, we have:

- Provided an intuition of the improved performance and sample efficiency of our method (Sec. 1).
- Further highlighted the differences between our approach and existing BO-related causal discovery methods (Secs. 1 and 2).
- Clarified our assumption regarding the low-rank structure of causal graphs (Sec. 4.1).

To strengthen our empirical evaluations, we have:

- Incorporated the source code for reproduction (Supplementary material). The demo code can be run very easily.
- Analyzed the benefits of low-rank representations in greater detail (Appendix F.1.6), suggesting that low-rank representations can lead to more diverse candidate DAGs and thus allow for reaching high-scoring DAGs earlier.
- Added more benchmark methods (GOLEM, NOTEARS, TMPI) to the main experiments (Sec. 5).
- Included conventional baselines (PC, GES) in supplementary experiments (Appendix G).
- Performed additional experiments on standardized data (Appendix F.1.5), demonstrating the robustness of our method in less-than-ideal scenarios.
- Conducted further performance and runtime analyses on large-scale nonlinear data, highlighting how our method efficiently achieves low SHDs even in high-dimensional and complex settings.

In light of these efforts, we kindly request your acknowledgment of our responses and, if deemed appropriate, a re-evaluation of our contribution. We eagerly look forward to your feedback and further discussion of our work before the end of the discussion period.

---

### Author Response · Authors · 2024-12-04
**Gratitude for your valuable feedback**

Dear Reviewers,

We would like to express our heartfelt gratitude for the time and effort you dedicated to reviewing our paper and contributing to a constructive and insightful discussion.

We are truly grateful that all reviewers agreed on a positive evaluation of our work and that the strong performance of our method has been verified. Your thoughtful comments and suggestions have also been invaluable in improving the quality of our manuscript, and we are committed to presenting an even more polished final version.

With your support, we are confident that our work will make a significant contribution to the causal discovery literature, offering a promising new direction for developing methods with enhanced accuracy and sample efficiency.

Once again, thank you for your dedication and support throughout this process.

Kind regards,

The Authors of DrBO

---

### Meta-Review · Area_Chair_5x2d · 2024-12-23

**Metareview:**

This paper introduces DrBO, a novel approach leveraging Bayesian Optimization (BO) for DAG learning. Overall, the reviewers praised the innovation and empirical results, particularly on dense graphs. In the rebuttal, the authors provided clarifications on runtime vs. number of DAG evaluations and added more baselines (e.g., GOLEM, NOTEARS-MLP, and TMPI). Concerns about assumptions (e.g., low-rank representations, identifiability) were addressed with explanations that DrBO’s setup is comparable to standard methods, and that the approach can handle both linear and nonlinear data. Despite questions about fairness in comparing continuous optimization methods, the authors demonstrated a consistent experimental design, including extended iteration budgets and runtime measures.

**Additional Comments On Reviewer Discussion:**

The reviewers actively participated in the discussion, and most concerns have now been addressed.

---

### Decision · Program_Chairs · 2025-01-22

Accept (Poster)